# Bridging Explainability and Embeddings: BEE Aware of Spuriousness

**Cristian Daniel Păduraru**[1,2]    **Antonio Barbalau**[1,2]    **Radu Filipescu**[1,2]

**Andrei Liviu Nicolicioiu**[3,4]    **Elena Burceanu**[1]

[1]Bitdefender, Romania
[2]University of Bucharest, Romania
[3]Mila, Montreal, Canada
[4]University of Montreal, Canada
`{cpaduraru, abarbalau, eburceanu}@bitdefender.com`
`andrei.nicolicioiu@mila.quebec`

## Abstract

Current methods for detecting spurious correlations rely on analyzing dataset statistics or error patterns, leaving many harmful shortcuts invisible when counterexamples are absent. We introduce **BEE** (Bridging Explainability and Embeddings), a framework that shifts the focus from model predictions to the weight space, and to the embedding geometry underlying decisions. By analyzing how fine-tuning perturbs pretrained representations, BEE uncovers spurious correlations that remain hidden from conventional evaluation pipelines. We use linear probing as a transparent diagnostic lens, revealing spurious features that not only persist after full fine-tuning but also transfer across diverse state-of-the-art models. Our experiments cover numerous datasets and domains: vision (Waterbirds, CelebA, ImageNet-1k), language (CivilComments, MIMIC-CXR medical notes), and multiple embedding families (CLIP, CLIP-DataComp.XL, mGTE, BLIP2, SigLIP2). BEE consistently exposes spurious correlations: from concepts that slash the ImageNet accuracy by up to 95%, to clinical shortcuts in MIMIC-CXR notes that induce dangerous false negatives. Together, these results position BEE as a general and principled tool for diagnosing spurious correlations in weight space, enabling principled dataset auditing and more trustworthy foundation models. Code publicly available HERE.

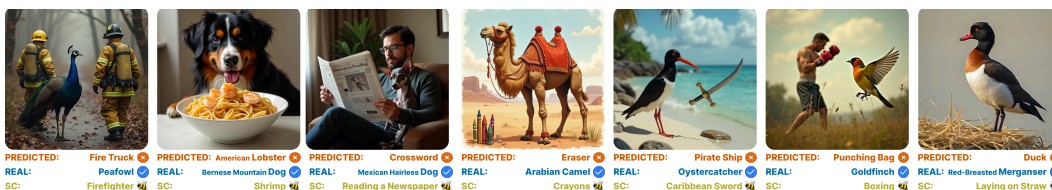

Figure 1: Qualitative results with BEE for CLIP ViT-L/14 fine-tuned on ImageNet-1k. Although a (REAL) class is clearly depicted, adding an object tied to a spurious concept (SC) flips the prediction to a (PREDICTED) class absent from the image, leading to unexpected and unwanted behavior.

## 1 Introduction and Background

Deep neural networks, and especially fine-tuned versions of foundation models, are increasingly deployed in critical areas such as healthcare, finance, and criminal justice, where decisions based on **spurious correlations (SCs)** can have severe societal consequences (Angwin et al., 2016; Caliskan et al., 2017). Even if a pretrained model has been validated by the community, the dataset leveraged

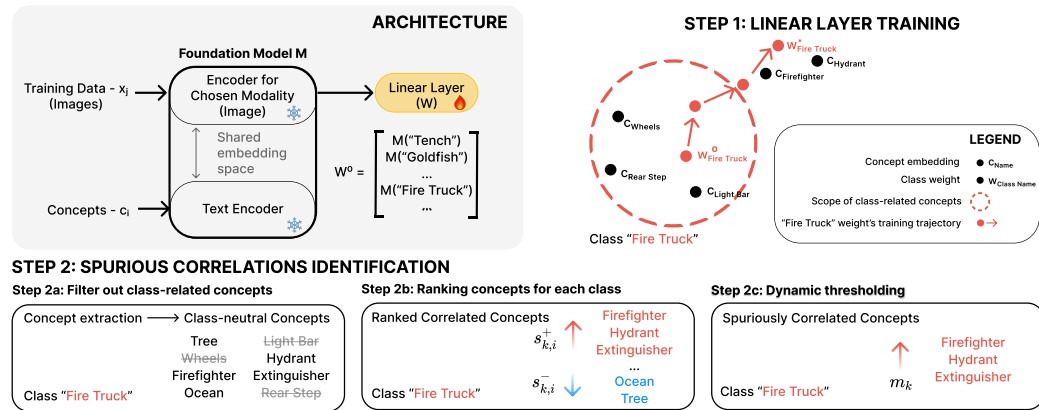

Figure 2: Following BEE's steps for the ImageNet-1k "Fire Truck" class. In Step 1, during training, the classification weights $W$ drift from the initial class concept embedding $W^0$, outside the scope of relevant concepts, towards spuriously correlated ones. In Step 2, our method filters out class-related concepts and, using an embedding-space scoring system, ranks and automatically marks the highest-ranking class-neutral concepts as SCs.

in the fine-tuning process can, and often does, imprint the model with new SCs. As shown in Fig. 1, a CLIP model fine-tuned on ImageNet-1k mislabels a *"Peafowl"* as a *"Fire Truck"* simply because a firefighter is present. This illustrates how fine-tuning can imprint hidden spuriousness in models.

**Limitations of existing approaches**   Research on *Spurious Correlations* has largely relied on two paradigms. *Data-driven methods*, such as SpLiCE (Bhalla et al., 2024) and Lg (Zhao et al., 2024), decompose data into high-level concepts and flag those disproportionately associated with certain classes. While useful for dataset auditing, these methods cannot determine which correlations are actually *learned* by the model, and may overlook low-frequency but impactful SCs. *Error-driven methods*, such as B2T (Kim et al., 2024b), infer SCs from validation errors. This require that held-out sets contain counterexamples that reveal the model's shortcuts, an assumption that holds in *subpopulation shift* benchmarks like Waterbirds (Wah et al., 2011b), CelebA (Liu et al., 2015a), and CivilComments (Borkan et al., 2019), but is rarely fulfilled in more general setups. To this end, Firca et al. (2025) highlight the way dataset splits impact the generalization capabilities of the model. Separating content from environment improves robustness, as spurious cues can dominate learned representations beyond what validation errors reveal Smeu et al. (2022; 2025). In contrast, our method identifies SCs learned during fine-tuning even without counterexamples, and it is meant to complement group-robustness methods (such as GroupDRO (Sagawa et al., 2020)) which require annotations for the dataset's SCs. We distinguish previously discussed works from methods for *SC-identification without external knowledge* (Liu et al., 2021a; Pezeshki et al., 2024; Zare & Nguyen, 2024), that do not explicitly *name* the spurious correlations driving decision errors and only partition datasets into easy and hard samples. The utility of their partitions is limited to improving group robustness within subpopulation shift setups alone. We also argue that without explicit identification, mitigation remains opaque, and the underlying vulnerabilities may persist unseen.

**Other partial alternatives**   *Counterfactual-image generation* (Singla & Feizi, 2021; Zhang et al., 2024a) probes fragility but depends on large generative models to correct the biases or costly super-vision. Another approach to detecting spurious correlations is to use models that are interpretable by design, such as *Concept Bottleneck Models (CBMs)* (Koh et al., 2020). These methods require a human expert to define the set of relevant concepts for each task, as well as providing a dataset with concept-level annotations in order to train the concept-extraction layer. To circumvent these limitations, Oikarinen et al. (2023) use concepts proposed by an LLM and obtain pseudo-labels for those concepts using a VLM. This intervention of CBMs on a models's architecture constrains its reasoning space down to the set of predetermined concepts, yielding drops in accuracy when compared to the unaltered models. Moreover, those models do not address the issue of SCs also being learned in the concept bottleneck layer, which questions their actual robustness in concept detection. Different from this line of works, we never constrain the model in any way, shape or form. What we

aim to uncover are SCs learned by general state-of-the-art models used in the industry, which are not explainable by design. Overall both approaches offer a different tradeoff between explainability and expressivity. A more detailed discussion of related work is provided in Appx. D.

**Our approach**  Prior work either focuses on mitigation without diagnosis, or on dataset-split level correlations without verifying what the models learn. In contrast, we target spurious correlations that models actually *learn*, even when no counterexamples exist in the training or validation sets to point out model misbehaviors. We introduce **BEE** (**B**ridging **E**xplainability and **E**mbeddings), a weight-space framework that tracks how fine-tuning perturbs pretrained representations. As shown in Fig. 2, classifier weights drift from their initial class embeddings toward spurious attributes. BEE exploits embedding geometry to rank class-neutral concepts outside the true class scope, exposing hidden drivers of biased decisions.

Our key contributions and findings are as follows:

**1. A weight-space diagnostic for spurious correlations.** We introduce **BEE**, the first framework to identify and *name* spurious correlations directly from the learned weights of a classifier. Unlike prior error-based or data-based methods, BEE uncovers correlations even when no counterexamples exist in training or validation splits.

**2. Evidence of persistence and transfer across models.** We show that SCs identified by BEE are not only artifacts of the studied classifier itself: they persist under full fine-tuning and transfer across diverse state-of-the-art backbones, reducing the ImageNet tested class accuracy by up to 95%.

**3. Broad, multimodal applicability.** We validate BEE across vision (Waterbirds, CelebA, ImageNet-1k) and language (CivilComments, MIMIC-CXR notes) tasks, as well as across multiple embedding families (CLIP, CLIP-DataComp.XL, mGTE, BLIP2, SigLIP2), proving relevance in a large scope of generic and realistic setups.

## 2 OUR METHOD

For a standard classification task, we aim to identify spurious correlations learned by a model through training on a new dataset. By **dataset's SCs** we refer to class-independent concepts, whose presence in the samples greatly affects the class label distribution. By **concepts** we refer to words or expressions with a well-defined semantic content. Through **SCs learned by a classifier** $f_\theta$ we refer to concepts causally unrelated with a class, whose presence in the input significantly changes the distribution of class probabilities predicted by $f_\theta$. We further classify them as positively correlated concepts w.r.t. to a class $k$, if their presence in the input increases the probability of $f_\theta$ predicting the class $k$, and negatively correlated concepts if they decrease it.

**Setup**  We start with a dataset of samples $(x_j, y_j) \in (\mathcal{X}, \mathcal{Y})$ and construct the set of concepts $c_i \in C_{all}$ in textual form, that are present in the training data. The details for building $C_{all}$ are provided in Sec. 3, Concept Extraction. We use a foundation model $M$, capable of embedding both the input samples $x_j$ and the concepts $c_i$ in aligned representations in $\mathbb{R}^D$.

The main steps of our method (also revealed in Fig. 2, and in more detail in the Alg. 2.1), are the following:

**Initialization**  We train a linear layer on top of the embedding space from $M$. We initialize $w_k$, the weights of class $k$ in this layer, with the embedding of its corresponding class name, as extracted by the model $M$:

$$w_k^0 = M(\text{class\_name}_k), k \in \overline{1, |K|}, \tag{1}$$

where $K$ is the list of class names.

**Step 1: Model training**  In the process of learning, the weights of each class $k$ in the linear layer naturally shift from their original zero-shot initialization, $M(\text{class\_name}_k)$, towards $w_k^*$. This drift is influenced by both class features and SCs, as illustrated in Fig. 2.

**Step 2: SCs identification** To identify the learned SCs, we exploit the fact that the weights $w_k$ and concept embeddings $M(c_i)$ share the same embedding space. To this end, we take the following steps:

**a. Filter out class-related concepts** After extracting the concepts $c_i$ present in the dataset using existing tools, we filter out the concepts that are related to any actual class. This leaves only concepts that are causally unrelated to all classes, which we call **class-neutral concepts**. We argue that only these concepts are proper candidates for SCs, as they are not required nor useful for the robust recognition of a class. For example, a forest background is well correlated with species of landbirds, but we want to prevent the model from relying on this correlation, which is unrelated to the class definition. The exact pipeline and tools for processing the concepts (extraction and filtering) are detailed in Sec. 3.

**b. Rank class-neutral concepts** For each class-neural concept $c_i$ and class $k$, we rank the concepts based on their similarities with the learned class weights $w_k^*$, using the following positive-SC score:

$$s_{k,i}^+ = w_k^{*\top} M(c_i) - \min_{k' \in \overline{1,|K|}} w_{k'}^{*\top} M(c_i). \tag{2}$$

Intuitively, we want concepts similar to one class but not to all the others. Thus, for each class $k$, we select the concepts which starkly correlate to it, compared to all other classes. For the negatively correlated SCs, $s_{k,i}^-$, we use the dissimilarity score: $-w_k^{*\top} M(c_i)$.

**c. Dynamic thresholding** We next keep only the highest-ranked SCs of each class, using a *dynamic threshold* for the scores above. This allows us to automatically select the SCs for each class. To smooth the curve of scores, we apply a mean filter with window size $r$ on top of the ranked concepts. We denote the scores obtained at this step with $(\overline{s}_{k,i})_{1 \le i \le p}$, with $p = q - r + 1$, where $q$ is the total number of filtered concepts. We then select the top $m_k$ ones, as positive SCs for class $k$, where the $m_k$ index is defined as:

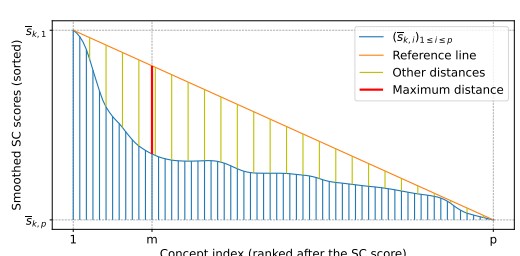

Figure 3: The maximum distance between the reference line and the smoothed scores gives the threshold for our cut-off heuristic.

$$m_k = \lfloor r/2 \rfloor + \arg\max_i \left( \overline{s}_{k,1} - i \frac{\overline{s}_{k,1} - \overline{s}_{k,p}}{p-1} - \overline{s}_{k,i} \right). \tag{3}$$

The intuition is that $m_k$ represents the index where the curve of smoothed scores, $\overline{s}_{k,i}$, deviates the most from the straight line connecting the points $(1, \overline{s}_{k,1})$ and $(p, \overline{s}_{k,p})$, as visually shown in Fig. 3.

## 2.1 ALGORITHM

We present the pseudocode of our proposed BEE approach in Alg. 1. We annotate the main steps presented in Sec. 2. At line 1, we initialize the class weights of our linear probing layer with the text embeddings of the class names, the zero-shot classification weights of the foundation model $M$. We then fine-tune the layer (line 2), on the given dataset. At lines 3-4, we filter the list of concepts and compute the embeddings for the remaining class-neutral concepts. The filtering can be performed by means of employing WordNet associations, Large Language Models, or both, as described in Sec. 3. With the embeddings of the class-neutral concepts at hand, we proceed to determine the SCs for each class. At line 5, we initialize our set of spuriously correlated concepts with an empty set, and we compute the similarities between each selected concept and each class at line 6. Next, for each class $k$, we compute the set of scores for all concepts w.r.t. this class and store it in $s_k$ (line 8). For each class-neutral concept, its ranking score for the current class is the difference between its similarity to the current class and the smallest similarity with a different class. Lines 9-11 formally implement the dynamic thresholding procedure, described in Sec. 2. Finally, we select the top concepts (above the computed threshold $m_k$) and store them as spuriously correlated concepts for class $k$ (lines 12-13).

---

**Algorithm 1** BEE - Weight space Approach to detecting learned SPuriousness

---

**Input**: M - foundation model with associated text encoder; $(\mathcal{X}, \mathcal{Y})$ - Training set; $(\mathcal{X}_{val}, \mathcal{Y}_{val})$ - Validation set; K - list of class names; $C_{all}$ - list of all concepts; r - window for dynamic thresholding.
**Output**: Identified positive SCs: $\mathcal{B}$.

  1: $\mathbf{W^0} \leftarrow M(K)$
  2: $\mathbf{W} \leftarrow \text{ERM}\left(\mathbf{W^0}, \mathbf{M}, (\mathcal{X}, \mathcal{Y}), (\mathcal{X}_{val}, \mathcal{Y}_{val})\right)$         ▷ 1. Model Training
  3: $C \leftarrow \text{Filter}(C_{all})$         ▷ 2a. Filter concepts: LLM/WordNet
  4: $\mathbf{C}^* \leftarrow \mathbf{M}(C)$
  5: $\mathcal{B} \leftarrow \emptyset$
  6: $\mathbf{S} \leftarrow \mathbf{W}^\top \mathbf{C}^*$         ▷ 2b. Rank class-neutral concepts
  7: **for** $k \in \overline{1, |K|}$ **do**
  8:     $s_k = \left[\mathbf{S}_{k,j} - \min_{k' \in \overline{1,|K|}} \mathbf{S}_{k',j}, \text{ for all } j \in \overline{1, |C|}\right]$     ▷ 2c. Dynamic thresholding
  9:     $\bar{s}_k = \text{mean\_pool}(\text{reversed}(\text{sorted}(s_k)), r)$
10:     $p = |C| - r + 1$
11:     $m_k = \lfloor r/2 \rfloor + \arg\max_i \left(\bar{s}_{k,1} - i\frac{\bar{s}_{k,1} - \bar{s}_{k,p}}{p-1} - \bar{s}_{k,i}\right)$
12:     $b_k = [s_{k,i} \mid i \leq m_k]$
13:     $\mathcal{B} = \mathcal{B} \cup (k, b_k)$         ▷ Positive SCs
14: **end for**

---

## 3 EXPERIMENTAL SETUP

**Foundation models (FM)** We used mGTE (gte-large-en-v1.5 (Zhang et al., 2024b)) for text embeddings in CivilComments, and OpenAI CLIP ViT-L/14 for text and images otherwise.

**Concept extraction** For the image classification task, we first use the GIT-Large (Wang et al., 2022) captioning model (trained on MSCOCO (Lin et al., 2014)) to obtain descriptions of the dataset's images. Next, we extract concepts from the captions (or directly from the text samples for the text classification task), using YAKE (Campos et al., 2020) keyword extractor, taking the top 256 n-grams for $n = 3, 5$. These keywords make up the set of concepts $C_{all}$.

**Concept filtering** We use Llama-3.1-8B-Instruct (Dubey et al., 2024) to remove class instances from the concepts extracted at the previous step. We also apply a post-processing based on Word-Net (Miller, 1995) to catch obvious class instances that the LLM might miss. For each class we specify a word used to search for synsets in WordNet (*e.g.* *bird* for Waterbirds) and then remove words that match with any of their hyponyms or hypernyms. The complete details are in Appx. F.

**Training** We train the linear layer on $L_2$ normalized embeddings extracted by the FM, using Py-Torch's (Paszke et al., 2017) AdamW (Loshchilov & Hutter, 2019) optimizer with a learning rate of $1e-4$, weight decay of $1e-5$ and batch size of 1024. We use the cross entropy loss with balanced class weights as the objective. The weights of the layer are normalized after each update and we use CLIP's temperature ($\tau = 100$) to scale the logits for each dataset and encoder. We use the validation set's class-balanced accuracy for model selection and early stopping. No explicit upper limit on the number of epochs is set. For the GroupDRO experiment in Section 4.5 we use $\eta = 1e-2$.

### 3.1 DATASETS

**Waterbirds** Sagawa et al. (2020) is a common dataset for generalization and mitigating spurious correlations.

It is created from CUB Wah et al. (2011a), by grouping species of birds into two categories, *landbirds* and *waterbirds*, each one being spuriously correlated with the background, land, and water respectively.

**CelebA** (Liu et al., 2015a) is a large-scale collection of celebrity images (over $200,000$), widely used in computer vision research. For generalization context, the setup Liu et al. (2015b) consists of using the *Blond_Hair* attribute as the class label and the *gender* as the spurious feature.

**CivilComments** (Borkan et al., 2019) is a large collection of 1.8 million online user comments. This dataset is used employed in NLP bias and fairness research concerning different social and ethnical groups.

**ImageNet-1k** (Deng et al., 2009) is a larger-scale popular dataset for image classification (1000 classes, with approx. 1300 training samples and 50 validation samples per class).

**MIMIC-CXR notes** (Johnson et al., 2019), a corpus comprising medical notes from chest examinations, for binary classification of the presence or absence of the pathological findings.

## 4 EVALUATING SPURIOUS CORRELATIONS

For a proper evaluation of our proposed SCs, we combine the concepts identified by **BEE** with different components.

In Sec. 4.1 we use SC-enhanced prompts for zero-shot classification with a Foundation Model, in some controlled setups, popular within the subpopulation shift literature. In Sec. 4.2-4.3 we expand to general, uncontrolled setups like ImageNet-1k and less explored ones like MIMIC-CXR medical notes. We further generate samples exploiting the discovered SCs here. In Sec. 4.4 we present some qualitative analysis on the SCs identified by BEE and competitors within the popular setups, underling their fundamental differences. In Sec. 4.5 we explore an extreme scenarios, lacking spurious correlation counterexamples. In Sec. 4.6 we further validate BEE on other embedding models. An extended list of the extracted concepts can be found in Appx. H.

Table 1: SC-enhanced zero-shot prompts. Following B2T, we inject SCs into zero-shot prompts, leveraging richer descriptions to improve classification. SCs identified by BEE significantly boost worst-group accuracy across image and text datasets.

| | Waterbirds (Acc % ↑) | | CelebA (Acc % ↑) | | CivilComm (Acc % ↑) | |
|---|---|---|---|---|---|---|
| Zero-shot | Worst | Avg. | Worst | Avg. | Worst | Avg. |
| Basic | 35.2 | 84.2 | 72.8 | 87.7 | 33.1 | 80.2 |
| w B2T | 48.1 | 86.1 | 72.8 | 88.0 | - | - |
| w SpLiCE | 48.1 | 82.5 | 67.2 | 90.2 | - | - |
| w Lg | 46.1 | 85.9 | 50.6 | 87.2 | - | - |
| w **BEE** | **50.3** | 86.3 | **73.1** | 85.7 | **53.2** | 71.0 |

### 4.1 SPURIOUS-AWARE ZERO-SHOT PROMPTING

To further validate our identified spurious correlations, we follow Kim et al. (2024b) and evaluate them in the context of a zero-shot classification task. We augment the initial, class-oriented prompt with the identified concepts through a *minimal* intervention (*e.g.* 'a photo of a {cls} in the {concept}' (see prompting details Appx. I). For each class we create a prompt with each identified spurious correlation. When classifying an image, we take into account only the highest similarity among the prompts of a class (zero-shot with max-pooling over prompts). We show in Tab. 1 how the SCs revealed by our method improve the worst group accuracy over the initial zero-shot baseline and other state-of-the-art solutions, in all the tested datasets. This highlights the relevance of the SCs automatically extracted by BEE. More ablation experiments can be found in Appx. J.

### 4.2 SPURIOUS CORRELATIONS IN IMAGENET-1K

Within this subsection, we apply our method in an uncontrolled, general setup. Specifically, we employ BEE to point out spurious correlations plaguing the decision-making process of OpenAI's CLIP ViT-L/14 fine-tuned on ImageNet. Within the ImageNet setup, the current state-of-the-art approach, B2T, points out the SCs learned by the model by analyzing the mistakes the model makes when evaluated on the validation set. Different from B2T, our approach does not rely on the validation data to provide counterexamples able to expose the SCs, and it is able to provide a list of SCs which exceeds the scope of the validation dataset. We provide extensive lists of SCs pointed out by our method in Appx. H. Most of the SCs pointed out by our method are previously untapped, opening up a new avenue for investigating ImageNet SCs.

Table 2: Results for three positively correlated SCs found using BEE for CLIP ViT-L/14 fine-tuned on ImageNet. We evaluate the model's capability to recognize a depicted (correct) class before and after the introduction of an identified concept in the image. For each prompt, 1000 images are generated using FLUX.1-dev. We observe throughout all considered scenarios, a significant drop in the model's capacity to identify the correct class when the selected concept is involved and a large increase in the likelihood of having the induced class predicted even though it is not illustrated.

| Correct Class | Exploited SC (Induced Class) | Prompt | Samples Predicted As (%) | |
|---|---|---|---|---|
| | | | Correct Class | Induced Class |
| **peafowl** | **firemen** (**fire truck**) | • a photo of a **peafowl** 
 • **firemen** and a **peafowl** | 100.0 
 5.3 (**-94.7**) | 0.0 
 93.4 (**+93.4**) |
| **Mexican hairless dog** | **reading a newspaper** (**crossword**) | • a photo of a **Mexican hairless dog** 
 • a man **reading a newspaper** in a chair with a **Mexican hairless dog** in his lap | 47.5 
 0.9 (**-46.6**) | 0.0 
 36.6 (**+36.6**) |
| **Bernese Mountain Dog** | **shrimp** (**American lobster**) | • a photo of a **Bernese Mountain Dog** 
 • **shrimp** and pasta near a **Bernese Mountain Dog** | 99.8 
 10.6 (**-89.2**) | 0.0 
 37.2 (**+37.2**) |

Table 3: Accuracy of various convolutional and transformer-based models trained on ImageNet-1k, on the data generated for Tab. 2. As with Fig. 1, we note that the performance of these models is significantly affected, even though the correct class is illustrated right in front and center while the predicted class is absent from the generated images. An exhaustive list is presented in Appx. K.

| Model | Prompt employed (correct class highlighted in **bold and blue**, SC in yellow) | | | |
|---|---|---|---|---|
| | a photo of a **peafowl** | firemen and a **peafowl** | a photo of a **Bernese Mountain Dog** | shrimp and pasta near a **Bernese Mountain Dog** |
| alexnet | 100.0 | 4.6 (**-95.4**) | 96.2 | 23.3 (**-72.9**) |
| efficientnet_b1 | 100.0 | 42.6 (**-57.4**) | 88.1 | 67.1 (**-21.0**) |
| regnet_x_32gf | 100.0 | 66.1 (**-33.9**) | 85.9 | 46.0 (**-39.9**) |
| resnet50 | 100.0 | 30.1 (**-69.9**) | 73.9 | 54.5 (**-19.4**) |
| resnext101_32x8d | 100.0 | 66.6 (**-33.4**) | 84.7 | 61.2 (**-23.5**) |
| squeezenet1_1 | 100.0 | 13.8 (**-86.2**) | 91.2 | 46.1 (**-45.1**) |
| swin_b | 100.0 | 81.5 (**-18.5**) | 95.2 | 72.6 (**-22.6**) |
| vgg19_bn | 100.0 | 35.9 (**-64.1**) | 83.1 | 46.2 (**-36.9**) |
| vit_l_16 | 100.0 | 55.9 (**-44.1**) | 95.3 | 76.0 (**-19.3**) |
| wide_resnet50_2 | 100.0 | 60.6 (**-39.4**) | 95.7 | 63.9 (**-31.8**) |

**Controlled SC validation via generative models** We further invest the effort to generate and manually verify images in order to open up this avenue and showcase previously undiscovered flaws in state-of-the-art models. To this end, we employ a quantized version of FLUX.1-dev (Labs, 2024), and in order to validate the impact of the SCs, we prompt the generative model to depict: (i) a chosen (correct) class, (ii) the same class alongside a SC (that is not an ImageNet class) that we found to induce other (induced) class, like the prompts shown in Tab. 2.

**Quantifying the impact of spurious concepts** The validation process is presented for three distinct scenarios in Tab. 2. Each scenario is defined by a correct class that is illustrated in the image, a concept (object, property, or activity) that is not causally tied to any class, and an absent class *induced* through the presence of the concept. We expect the classifier to predict this absent class, based on our scoring. We measure the impact of the concept by comparing the model's ability to predict the correct class before and after its introduction. We generate and manually ensure the compliance of 1000 images for each scenario and we evaluate the model both in terms of accuracy and in terms of the frequency with which it predicts the induced class. Throughout all considered scenarios, we observe a significant drop in the model's capacity to identify the correct class when the SC factor is involved, with an increased likelihood of having the induced class predicted, even though it is not illustrated in any way, shape or form in the image. We present smaller-scale tests for alternatives to the generative model FLUX, yielding similar results, in Appx. C.

Table 4: Qualitative SCs examples, extracted on Waterbirds, CelebA and CivilComments datasets. See in orange concepts that are off-topic, person names, or too related to the semantic content of the class, and in blue new concepts, that were not identified before. BEE, w.r.t. others, focus on learned SCs, discovering many new spuriously correlated concepts (and expressions, marked with ...).

| | Waterbirds | | CelebA | | CivilComments | |
| | *landbird* | *waterbird* | *blonde hair* | *non-blonde hair* | *offensive* | *non-offensive* |
|---|---|---|---|---|---|---|
| B2T | forest, woods, tree, branch | ocean, beach, surfer, boat, dock, water, lake | - | man, male | - | - |
| SpLiCE | bamboo, perched, rainforest | flying | hairstyles, dolly, turban, actress, tennis, beard | hairstyles, visor, amy, kate, fielder, cuff, rapper, cyclist | - | - |
| Lg | forest, woods, rainforest, tree branch, tree | beach, lake, water, seagull, pond | ...blonde hair, actress, model, woman long hair | man..., sunglasses, young man, black hair, actor | - | - |
| **BEE** (ours) | forest..., bamboo..., ground, field, log, grass..., tree | swimming..., water, lake, flying..., boat, lifeguard, pond | - | hat..., man..., actor, person, dark, large, shirt | hypocrisy, troll, solly, hate | allowing, work, made, talk |

**Qualitative failures in ImageNet-1k** Within the same context, we present a series of qualitative examples in Fig. 1. We emphasize that, even though throughout most of these samples, a single ImageNet-1k class is clearly depicted, the model chooses to ignore it and label the image as a completely different class, not illustrated at all in the image, solely based on the presence of a non-ImageNet object. We underline, by means of the results presented in Tab. 2, that the model is not fooled by artifacts in the generated images to predict randomly. We test the performance of the models on images featuring the correct class, without added objects. We observe this way that the model's performance on the generated data is on par with the original performance of the model on these classes, validating that the generated images are not out of distribution. Furthermore, we show that the rate at which the induced class is predicted increases significantly.

**Generalization across state-of-the-art models.** Although the analyzed model is a strong, well-pretrained baseline, our experiments show that critical reasoning flaws can persist in production-ready models and remain undetected by standard held-out evaluation. We further evaluate the identified SCs on a broad set of ImageNet-1k state-of-the-art models, with full results in Appx. K and selected results in Tab. 3. Notably, even large transformer models such as ViT-L/16 are strongly affected by learned SCs, highlighting their generality and silent impact across modern architectures.

## 4.3 Spurious Correlations in MIMIC-CXR Notes

To highlight the practical value of our approach and demonstrate BEE's utility in more specialized domains, we employ it on the clinical notes from the MIMIC-CXR dataset, using the *"no finding"* label as the classification target. BEE revealed concepts such as *"chest examination"* and *"chest radiograph"* as being spuriously correlated with the *"no finding"* class. We find this to match with patterns in the training data: *"finding"* samples mention pathologies explicitly, whereas *"no finding"* samples often reference the examinations as showing no issues.

**Adversarial example** Given the sentence *"The chest examination found signs of disease and the chest radiograph exam found the same"*, our mGTE-based classifier incorrectly predicts it as *"no finding"*, a serious error despite the explicit mention of disease.

**Quantitative validation** Adding the phrase *"chest examination"* to all samples does not change their label for the considered task, and yet leads to predictions more biased towards the *"no finding"* class, increasing its recall by 2.1%, while decreasing the recall of the *"finding"* class by 2.2%.

## 4.4 Qualitative examples

We present in Tab. 4 the concepts identified as spuriously correlated with each class by BEE and competitors. Notice how our method discovers many new concepts (in blue) when compared with

Table 5: Learning in the context of perfect spurious correlations. In the absence of samples that associate a class instance and concepts spuriously correlated with other classes, GroupDRO does not outperform the standard ERM. In contrast, our regularization based on the identified concepts consistently yields improvements (concerning worst group accuracy) over the considered baselines: ERM, GroupDRO, and the regularization with random causally unrelated concepts (obtained after the filtering in Step2a).

| Method | Waterbirds (Acc % ↑) | | CelebA (Acc % ↑) | | CivilComments (Acc % ↑) | |
|---|---|---|---|---|---|---|
| | Worst | Avg. | Worst | Avg. | Worst | Avg. |
| ERM | 43.2±5.7 | 72.7±2.2 | 9.6±1.0 | 58.2±0.4 | 18.6±0.3 | 49.9±0.2 |
| GroupDRO | 38.9±5.4 | 71.2±2.0 | 8.1±0.3 | 60.3±1.0 | 18.7±0.4 | 50.2±0.5 |
| Regularize w/ random SCs | 46.6±2.7 | 75.3±1.1 | 9.4±0.0 | 61.4±2.0 | 19.1±1.6 | 50.8±0.9 |
| Regularize w/ Lg's SCs | 50.4±0.1 | 76.6±0.0 | 8.3±0.0 | 61.2±0.5 | - | - |
| Regularize w/ **BEE**'s SCs | **57.9**±0.3 | 79.8±0.1 | **10.4**±0.5 | 62.0±1.8 | **31.3**±0.7 | 57.5±0.4 |

others. This is because our approach is fundamentally different, as it relies on the decision-making process of the model being investigated, diverging from current techniques oriented to validation set errors (B2T), or others that do data analysis over frozen concepts (SpLiCE, Lg). For CelebA-*blonde hair*, B2T and BEE do not find any SCs. This turn out to be an appropriate decision, since the presence of the feminine features do not incline the model towards one class or the other. See an exhaustive list of SCs revealed by BEE (ImageNet-1k included) in Appx. H.

## 4.5 TRAINING IN A FULLY SPURIOUS SETUP

We consider an extreme setting with no spurious-correlation counterexamples by removing minority groups from standard robustness datasets (e.g., waterbirds on land). In this regime, GroupDRO-style methods are ineffective, as no underrepresented groups remain, matching ERM performance at best (Tab. 5). Such correlations can naturally arise from dataset acquisition choices. To improve robustness, we regularize linear probes using the identified SCs by constraining class weights to be equally distant from them, formulated as an MSE between each class weight's similarity to an SC and the average similarity across classes, scaled by CLIP's temperature $\tau$.

$$\mathcal{L}_{reg}(b) = \frac{\tau^2}{N} \sum_{k=1}^{N} \left[ w_k^\top M(b) - sg \left( \frac{1}{N} \sum_{j=1}^{N} w_j^\top M(b) \right) \right]^2, \qquad (4)$$

with *sg* being the stop gradient operator. The final loss is $\mathcal{L} = \mathcal{L}_{ERM} + \alpha \frac{1}{|\mathcal{B}|} \sum_{b \in \mathcal{B}} \mathcal{L}_{reg}(b)$, where $\mathcal{B}$ is the set of selected concepts and $\alpha = 0.1$. Tab. 5 reports linear probing results in the no-counterexample setting. SC regularization reduces reliance on revealed SCs and improves worst-group accuracy, indicating more robust and better-aligned classification weights. For comparison, we replicate Lg's SC-identification process in this scenario.

## 4.6 CROSS-MODEL VALIDATION

We further validate BEE on other embedding models. In addition to CLIP ViT-L/14, we also used CLIP ViT-L/14 DataCompXL, BLIP2, and SigLIP on the Waterbirds dataset. Using SCs identified by BEE, based on embeddings from each of these models, in the zero-shot setting (Tab. 6) led to a significant improvement in worst-group accuracy (while the average accuracy stays in the same range). This highlights BEE's effectiveness across diverse embedding models.

Table 6: Zero-shot Waterbirds results with different embeddings. BEE proves to be robust, improving worst-group accuracy across models, while keeping average accuracy stable.

| Zero-shot | Worst Acc. % | Avg. Acc. % |
|---|---|---|
| Basic CLIP | 35.2 | 84.2 |
| w BEE-CLIP | 50.3 | 86.3 |
| w BEE-DataComp | 45.5 | 85.6 |
| w BEE-BLIP2 | 54.8 | 86.0 |
| w BEE-SigLIP2 | 49.7 | 85.6 |

### 4.7 HARDWARE AND COMPUTE TIME

We used an NVIDIA RTX 4090 for captioning and image generation and an NVIDIA RTX 2080 for linear probing. For ImageNet-1k, the preprocessing takes 12 hours, making it the most expensive dataset to preprocess. This is incurred only once per dataset. Afterward, models can be evaluated efficiently, with linear probing taking $\sim 2$ minutes per encoder.

## 5 DISCUSSION: INSIGHTS AND LIMITATIONS

Below we outline several limitations of BEE, along with observations that help contextualize its applicability and scope (more in Appx. E):

**Spurious Correlations: Dataset-Induced or Foundation-Model-Induced?** Our empirical results suggest that the spurious correlations surfaced by BEE arise more from the downstream dataset than from the foundation model, though formally disentangling these factors remains an open challenge. BEE does not aim to separate these bias sources, but to reveal the shortcuts the fine-tuned model actually relies on. *Empirical evidence*. On datasets with known spurious attributes (e.g., CELEBA, WATERBIRDS), BEE reliably recovered the expected shortcuts, indicating strong dataset-level effects. On IMAGENET, the concepts identified by BEE degraded performance across architectures trained independently of CLIP (e.g., EfficientNet, ResNet, ViT; see Tab. 3). This suggests that these correlations are learned during downstream training rather than inherited from frozen embeddings.

**Are Linear Probes Sufficient for Spurious Correlation Analysis?** At first, BEE approach appear to be limited to linear or near-linear relationships. However, this reliance on linear probing is grounded in findings showing that such relationships are often sufficient for diagnosing spuriousness in the final representation layer. Prior work (Kirichenko et al., 2023) shows that spurious and core features tend to be linearly separable in the final representation layer, supporting our methodological choice. Additionally, recent findings using Sparse Autoencoders (Huben et al., 2024; Barbalau et al., 2025) demonstrate that many meaningful and disentangled features can be isolated within a single linear layer. Based on these findings, linear probing offers an effective balance between expressiveness and interpretability for diagnosing spurious correlations.

**Broader impact of BEE-SCs (apart from classification)** The spurious correlations we detect often reflect co-occurrences inherently present in the dataset. For example, in ImageNet, firefighters frequently appear together with fire trucks. Such co-occurrences are task-agnostic and remain in the data regardless of the specific downstream objective. This means that any model trained on the same dataset, whether for classification, VQA, or another task, can implicitly learn these associations. For instance, a VQA model trained on such data might learn to answer *yes* when asked whether a fire truck is present whenever it sees a firefighter, even if none is shown. Therefore, what BEE reveals are root causes of shortcut learning that can propagate across tasks built on the same dataset. Although their exact effect depends on the task, these patterns represent general dataset-level dependencies that are valuable to detect and understand beyond classification. Although we did not evaluate these spurious correlations on other tasks, their dataset-level nature means they could transfer to models trained on the same data, highlighting a promising avenue for future work.

## 6 CONCLUSIONS

We introduced **BEE**, a weight-space framework for detecting and naming spurious correlations without relying on counterexamples. Our experiments across vision, language, and medicine show that the correlations uncovered by BEE persist across full fine-tuning and the method is generic for diverse foundation models. By exposing hidden shortcuts with interpretable signals, BEE complements existing mitigation methods and provides a practical tool for dataset auditing and building more trustworthy systems.

### ACKNOWLEDGEMENTS

The project was funded by the EU Horizon project ELIAS (No. 101120237).

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

APPENDIX

## A  BROADER IMPACT STATEMENT

By systematically detecting a wide spectrum of spuriously correlated concepts, our work stands to enhance the reliability and trustworthiness of AI-driven decisions across various real-world contexts. BEE could help researchers and developers to address unintended consequences that arise when models latch onto misleading data associations, drawing attention to critical responsibilities tied to deploying AI at scale.

## B  SOFTWARE AND DATA

We make our BEE code in PyTorch Paszke et al. (2017) publicly available HERE. The datasets and pre-trained models used for BEE are already public.

## C  ALTERNATIVES TO THE GENERATIVE MODEL

**ChatGPT-4o generator.**  We performed a small-scale experiment with an alternative generative model, other than FLUX.1-dev from the main paper. Forty images were generated with ChatGPT-4o (OpenAI, 2023): half depicted only a peafowl, and half depicted both a peafowl and a firefighter, but no firetruck. The linear probe trained on top of CLIP achieved 100% accuracy on the first set, yet its accuracy dropped to 50% once the SC was introduced. This decline confirms that the evaluation results are not intrinsically dependent on the FLUX generative model. Furthermore, in the cases where the model failed, it was always because it predicted the fire truck class, though no fire truck was present in the image, while the peafowl was realistically depicted and occupied a large portion of the image. The fact that the model achieves 100% accuracy when the SC is not involved, and the fact that it proceeds to misclassify the input image into the specific class suggested by the SC, suggest that these errors are not due to artefacts in the generative models, but rather due to actual flaws of the investigated model.

**Natural images.**  If we avoid synthetic data altogether, scaling the evaluation becomes impractical: each image must be located, licensed, and annotated by hand. To probe the phenomenon nevertheless, we performed a focused Google News search and selected naturally occurring photos for the same SC. Our investigation of real images confirmed the same statistics presented for the ChatGPT-4o generator.

## D  DETAILED RELATED WORK

Machine learning methods naturally capture relevant factors needed to solve a task. However, models might also capture shortcuts (Geirhos et al., 2020), as correlations between non-essential features of the inputs and the label. These shortcuts represent spurious correlations, that don't hold in a more general setup (*e.g.* using water features to classify waterbirds instead of focusing on the birds' features), and should not be used for reliable generalization outside the training distribution, as they often lead to degraded performance (Quiñonero-Candela et al., 2008; Beery et al., 2018; Hendrycks et al., 2021).

**SCs from error analysis**  Approaches like **B2T** Kim et al. (2024b) rely exclusively on the validation samples, identifying which correlations between concepts are more prevalent in the misclassified examples. To catch SCs, B2T needs samples that oppose the strong correlations in the training set, thus leading to misclassification. To circumvent this limitation, **DrML** Zhang et al. (2023b) manually builds a list of texts containing classes and concepts associations, that could potentially underline an SC. It forwards each such textual association through a classifier, learned on the image modality, and keeps as SCs the erroneously classified ones. In those approaches, the burden falls on the practitioners to come up with exhaustive samples or associations, failing to detect unexpected SCs. Differently, **BEE** focuses on analyzing the explicit learned weights of a model, covering

all trainset samples. We thus extract the spuriously correlated concepts directly from the weights, bypassing the need for an exhaustive validation set or correlations candidates.

**SCs from train data analysis**   In **SpLiCE** (Bhalla et al., 2024) each image is decomposed into high-level textual concepts, searching next for concepts that are frequent for a certain class, but not for the others. **LG** Zhao et al. (2024) relies on LLMs to propose concepts potentially correlated with each class, using image captions. Next, it uses CLIP (Radford et al., 2021) to estimate a class-specificity score for each concept, and highly scored concepts for a class w.r.t. the others are considered SCs. These methods focus on the concepts' occurrences per class, making them prone to missing low-frequency concepts, as their presence can be drowned when averaging scores over a large dataset. Moreover, the SCs found through data analysis could be harder to learn than the class itself, so they are not necessarily imprinted in the model. In contrast, learned SCs (including error analysis revealed ones) must always be addressed, as they are, by definition, proven to impact the classifier. For this reason, **BEE** targets learned SCs by looking directly at the impact of the training set upon the model's weights.

**Manual interpretation of correlations**   The method introduced in Singla & Feizi (2021) finds spurious features learned by a model, but it requires humans to manually annotate whether an image region is causal or not for a class. While this ensures a higher quality of the annotations, it also poses problems of scalability to large datasets. In contrast, **BEE** works fully automated, at scale, identifying SCs for each class in ImageNet-1k.

**SCs from subpopulation shift setup**   Other previous works (Pezeshki et al., 2024; Liu et al., 2021a; Arefin et al., 2024; Zare & Nguyen, 2024) have focused on SC identification strictly within the context of subpopulation shifts. The particularity of this setup is that the training and validation sets always contain subsets of samples that oppose the strong spurious correlations of the dataset. Most of these methods (Pezeshki et al., 2024; Liu et al., 2021a; Zare & Nguyen, 2024) focus on first learning a strongly biased classifier and then either separate the samples of each class into two groups (Pezeshki et al., 2024; Zare & Nguyen, 2024) (one containing correctly classified examples and the other containing misclassified ones), or place higher weights on hard samples (Liu et al., 2021a), in order to balance the dataset. **CoBalT** Arefin et al. (2024) on the other hand uses an unsupervised method for object recognition and then samples the dataset examples such that all object types are uniformly distributed in each class. Their result contains heatmaps overlays on images, which can offer insights to guide further manual SCs identification. Some of the most commonly used datasets in this setup are Waterbirds (Sagawa et al., 2020), CelebA (Liu et al., 2015a) and CivilComments (Borkan et al., 2019).

**Preventing the learning of SCs**   As the statistical correlation of attributes and classes lies at the root of learning SCs, breaking this correlation is an accessible way of preventing their learning. Assuming that the training set features all combination of classes and attributes, this can be achieved by balancing all the existing groups of samples, as defined by the intersection of class and attribute labels. **GroupDRO** (Sagawa et al., 2020) uses group-specific weights that are dynamically updated during training to balance them, and it is the approach most commonly taken by works that simply find dataset partitions (Pezeshki et al., 2024; Zare & Nguyen, 2024), and also by works that name the SCs, and then obtain pseudo-labels for those attributes (Kim et al., 2024b; Zhao et al., 2024). To judge the robustness of a classifier, the **worst-group accuracy** metric is employed, which computes the accuracy on each individual group of samples and then reports their minimum. The worst-group accuracy of GroupDRO with ground truth attribute labels is usually viewed by the previously mentioned works as an upper bound on the performance that can be obtained.

**Fairness**   It is important to note that the proposed method can be utilized to evaluate the fairness of a given dataset and that we do conduct benchmarking on the CivilComments dataset, which encompasses racial and religious concerns. However, it is critical to emphasize that our approach is neither designed to measure nor address issues of fairness. Instead, our method is specifically developed to examine whether a given dataset imparts a clear definition of the featured classes to a model, namely, whether classifiers learn spurious correlations and confound class features with environmental features. Accordingly, our work is situated within the literature on subpopulation shift setups and we assess the quality of our proposed approach within this framework. Evaluating

our approach on fairness benchmarks lies outside the scope of the current study, but may constitute a subject for subsequent research.

## D.1 CONCEPT BOTTLENECK MODELS

Another approach to detecting spurious correlations would be to use models that are interpretable by design, such as Concept Bottleneck Models (CBMs) (Koh et al., 2020). CBMs feature a special layer where each neuron's activations signals the presence or absence of a specific concept within the input sample. This makes it easier to see which concepts are used by the model down the line and also allows a user to filter out the concepts that he may consider as irrelevant for the task at hand. On the downside, CBMs, as proposed by Koh et al. (2020), require a human expert to define the set of relevant concepts for each task and also concept-level annotations in a dataset in order to train the concept extraction layer. To circumvent these limitation, Oikarinen et al. (2023) use concepts proposed by GPT-3 and then obtain pseudo-labels for those concepts using a CLIP model. This intervention of CBMs on a models's architecture constrains its reasoning space down to the set of predetermined concepts, yielding, compared to unaltered models, drops in accuracy of up to 4.97%, as reported by Oikarinen et al. (2023) in Table 2. Different from this line of works, we never constrain the model in any way, shape or form. What we aim to uncover are SCs learned by general state-of-the-art models used in the industry, which are not explainable by design. Overall both approaches offer a different tradeoff between explainability and expressivity.

**Oikarinen et al. (2023)** Similar to our approach, the method proposed by Oikarinen et al. (2023) can be applied in general setups, but with some caveats. They train a concept layer, within which each neuron signals the presence or absence of a predetermined concept proposed by GPT-3. This intervention constrains the reasoning space of the model down to the set of predetermined concepts, yielding, compared to unaltered models, drops in accuracy of up to $4.97\%$, as reported in Table 2 from Oikarinen et al. (2023). Different from them, we never constrain the model in any way, shape or form. Our analysis is completely post-hoc, allowing the model to reason in its natural embedding space while achieving the best performance it can on the given dataset.

**Yüksekgönül et al. (2023)** Similar to our method, the approach proposed by Yüksekgönül et al. (2023) is post-hoc. That is, the authors train a CBM layer on top of a pre-trained non-CBM model. Furthermore, their approach features a mechanism designed to reduce the performance loss usually seen in CBM models. In this respect their work differs significantly from prior CBM literature. Our work differs from this approach in two main respects pointed out directly by Yüksekgönül et al. (2023) in their *Limitations and Conclusion* section of their paper: *Users should be careful about the concept dataset used to learn concepts, which can reflect various biases. While there are several such real-world tasks, it is an open question if human-constructed concept bottlenecks can solve larger-scale tasks(e.g. ImageNet level).* First, in this direct quote, the authors point out that their method cannot find out the spurious correlations learned by their bottleneck layer during training. In contrast, our method is specifically designed to point out the spurious correlations learned by our classifier during training. Second, while in their work, the possibility of applying their method to large-scale datasets such as ImageNet remained an open question, we have successfully applied our method on the ImageNet dataset.

**Kim et al. (2024a)** This work further improves upon the method proposed by Yüksekgönül et al. (2023) by means of introducing a mechanism meant to automatically obtain unbiased MLLM attribute annotations in order to train the bottleneck layer without human supervision. In terms of differences, even if the MLLM annotations were to match the human level, the limitations pointed out by Yüksekgönül et al. (2023) still remain. That is, the spurious correlations learned by the bottleneck layer itself during training remain hidden. The annotation procedure ensures that the labels that are used are as accurate as possible, but it cannot determine what spurious correlations are present in the training dataset used for the bottleneck layer. In contrast, our method investigates the spurious correlations learned by our classifier during training.

## E    DISCUSSION: INSIGHTS AND LIMITATIONS (CONTINUATION)

**BEE's reliance on external models (concept extraction, LLM and WordNet filtering)** These components are used only for candidate generation and filtering, so their impact on the core di-

agnostic signal is limited. Potential biases could cause either *excessively-* or *insufficient-filtering*: *Over-filtering* might remove valid spurious cues, yielding too few or trivial SCs, but this was not observed, BEE consistently surfaced novel and meaningful shortcuts (see ImageNet results). *Under-filtering* might admit class-related concepts and incorrectly flag them as spurious; however, our experiments did not show this effect at scale. Incorporating BEE-identified SCs consistently reduced model robustness (Tab. 2-3), confirming their spurious nature. *Overall*, these heuristic steps support scalability, while the core diagnostic signal comes from the geometric alignment between class weights and concept embeddings, which is independent of auxiliary model biases.

**Concepts vs Input features as SCs** The learned SCs can be described by our method in relation with the predefined (large set of) concepts, but not directly w.r.t. the input features (*e.g.* Grad-CAM (Selvaraju et al., 2017) like methods).

**Trade-off between accuracy and fairness** Removing SC reliance often reduces average accuracy, since SCs are predictive within biased datasets. This drop is widely recognized in the robustness literature and we report both average and worst-group metrics to transparently reflect this trade-off.

**Generalizability** While our mitigation experiments focus on binary classification benchmarks, BEE detection itself is agnostic to task cardinality, as evidenced by the ImageNet experiments.

**Captioning model used for extracting concepts** These models usually do not extract all the details in the images, so relying on them limits the concept space, that limits further discovering all SCs from the original images.

**SCs from a dataset (only) through the lens of a Foundation Model** While the Foundation Models are usually very robust ones, some SCs (specially those related to low-level- pixel-level - information) can disappear in the high-semantic embedding space of the foundation model, making it impossible for BEE to detect such SCs.

**Relying on known hierarchies of concepts** The method also relies on known hierarchies of concepts (like WordNet) to filter out concepts related to the desired class. These hierarchies and the relations they provide thus limit the type of filtering that we can ensure.

# F  CONCEPT FILTERING RULES

In the filtering stage we process a list of n-grams extracted from the set of image captions or from the set of samples in the case of text dataset, and output a list of strings that do not contain concepts related to the classes.

## F.1  LLM-BASED FILTERING

We provide below the prompt used for filtering the list of concepts.

---

**Prompt used for concept filtering with LLMs**

I will provide a list of concepts and sequence of words. Your task is to remove any instance of the concepts from the given sequence. If no instance of any concept is present then you must return the sequence as is.

Here are a few examples:

Example 1:
Concepts: [dogs and any specific species of dogs]
Sequence: 'a golden retriever with a bone'
Answer: 'bone'

Example 2: Concepts: [clothing and anything related to their color]
Sequence: 'a shiny black and white dress'
Answer: 'shiny'

Example 3:
Concepts: [mentions of peopleś names]
Sequence: 'John is an assistant'
Answer: 'assistant'

Example 4:
Concepts: [cats, horses, dolls, the sun and any specific species or types of these concepts]
Sequence: 'A picture of the rising sun'
Answer: 'picture'

Now complete the following case, without thinking step by step or asking for anything else.
Concepts: [{}]
Sequence: '{}'
Answer:

---

We next describe the rules used to process the response of the LLM:

- We first locate the two apostrophes that are expected to be in the answer and keep only the part of the answer that lies between them. A null string results if two apostrophes are not present in the answer.
- We ignore an n-gram if the answer of the LLM is a null string or if it contains words not in the initial sequence. This can signal a hallucination or failure to adhere to the required output format. We used the word tokenizer from nltk in this step.
- For n-grams where the LLM provides a valid answer, we trim the starting and trailing stopwords. We use the set of english stopwords in nltk, as well as the words "next" and "many", which we found to be often produced by the chosen captioning model.
- If the answer of the LLM only contained stopwords we discard it. Otherwise it is added to the list of candidates.

## F.2 WORDNET-BASED FILTERING

For this approach we use the folowing rules:

- We first tokenize each n-gram into individual words with the word tokenizer in nltk.
- We then lemmatize each word and mark it as class-related if its lemma appears in the class name (e.g., "blonde" appears in "blonde hair" for CelebA) or if it is a hypernym or hyponym of the corresponding WordNet synsets for any class (e.g., "pelican" is a hyponym of "bird"). Note that the user could define one or more WordNet synsets for a class if it is a union of multiple synsets in WordNet, and a fitting hypernym does not exist. E.g., class 0 in a dataset could represent venomous animals, which includes certain snake species, as well as spiders, stingrays etc..

- Each n-gram with no class-related words is added in the final list of candidates
- For the other n-grams we remove all class-related words. If only stopwords remain, we discard them. Otherwise, the resulting string is added to the list of candidates.

At the end of the filtering stage we deduplicate the obtained list of candidates. To be noted, we can have a word with both its singular and plural form in the list of candidates. The lemmatization in WordNet-based filtering is only used to determine if the word is related to the classes or not, but raw words are used in subsequent steps. This is an intended behavior, because if only the plural form ever appears then we could gain additional information about the dataset - the entity always appears in a group.

## G  Loss correlation with presence of spuriously correlated concepts

In this experiment, we look at the correlations between the loss values and the concept-to-sample similarities. We compare basic ERM with GroupDRO, applied on groups, that are obtained based on our revealed SCs (and further grouped using the B2T Kim et al. (2024b) partitioning strategy).

See in Fig. 4 how for GroupDRO, the loss-to-similarity correlation significantly decrease, revealing that the model is less prone to make mistakes on the samples containing SCs. The results show a reduction in correlation scores across all SCs, demonstrating that the revealed groups are relevant to the dataset's underlying distribution, and can be effectively utilized with specific algorithms to mitigate the model's dependence on spurious correlations. Fig. 4 shows the Pearson correlation scores after an epoch of training on Waterbirds, on a subset of all SCs.

## H  SCs and top class-neutral concepts

We present the exhaustive list of SCs found for the Waterbirds (Tab. 15 & 16), CelebA (Tab. 17) and CivilComments (Tab. 18 & 19) datasets. We also present top class-neutral concepts for ImageNet classes. Concept filtering on ImageNet was performed using WordNet relationships alone, without the intervention of a Large Language Model. Accounting for the size of the dataset, we will publish the available data on our repository upon acceptance, and we will restrict the presentation within the context of the current format to a few classes for illustrative purposes in Tab. 28 -21.

**Number of SC candidates** After passing through both filtering stages (LLM-based and WordNet-based), we were left with the following number of candidates per dataset:

- Waterbirds: 294
- CelebA: 280

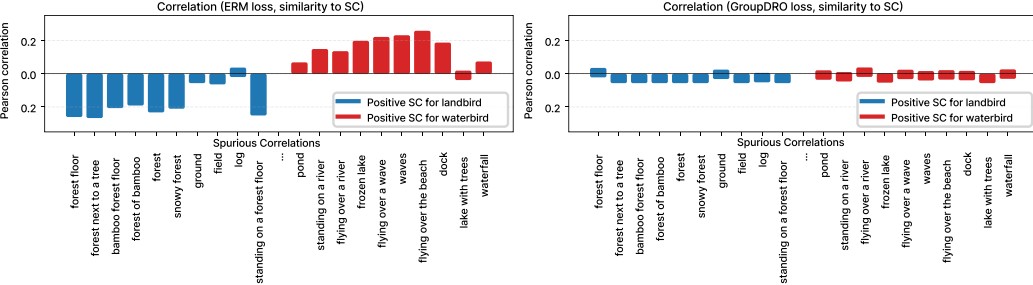

Figure 4: Correlation(sample_loss, sample_to_bias similarity) under ERM/GDRO after one epoch of training on Waterbirds. Loss correlation w/ biases, ERM vs GroupDRO using groups created with the B2T partitioning method. It can be seen that, when training with ERM, loss value is highly correlated with the biases. In contrast, GroupDRO reduces the correlations, intuitively showing that biases discovered with our method are closely related to the ground truth groups of the dataset, being used as shortcuts by the model unless mitigated.

Table 7: Class names and prompts used in the zero-shot classification task.

|  | Waterbirds | CelebA | CivilComments |
|---|---|---|---|
| class names | waterbird landbird | non-blonde hair blonde hair | non-offensive offensive |
| zero-shot prompt | a photo of a {cls} | a photo of a person with {cls} | {cls} |
| SC-enhanced prompt | a photo of a {cls} in the {SC} | a photo of a {SC} with {cls} | a/an {cls} comment about {SC} |

Table 8: Ablation. Following the zero-shot SC-augmented prompting setup, we variate the cut-off threshold for considering causally-unrelated concepts as spuriously correlated and also try to address the text-image modality gap.

| Variations | Waterbirds (Acc % ↑) Worst | Avg. | CelebA (Acc % ↑) Worst | Avg. | CivilComm (Acc % ↑) Worst | Avg. |
|---|---|---|---|---|---|---|
| *top-30 candidates* | 46.3 | 86.1 | 66.2 | 86.4 | 46.9 | 71.1 |
| *top-20% candidates* | 46.1 | 86.0 | 64.8 | 86.7 | 48.8 | 66.2 |
| *modality gap: closed* | 48.7 | 85.9 | 72.7 | 85.3 | - | - |
| **BEE** *\* dynamic threshold* *\* modality gap: open* | **50.3** | 86.3 | **73.1** | 85.7 | **53.2** | 71.0 |

- CivilComments: 587

- ImageNet: 368,170

# I  ZERO-SHOT PROMPTS

In Tab. 7, we structured the class names used for initializing the initial weights of the linear layer, along with the prompt templates employed in the zero-shot classification experiments discussed in Sec. 4.1.

# J  ABLATIONS

## J.1  THRESHOLDING STRATEGIES

We validate several BEE decisions in Tab. 8, for zero-shot classification task, using prompts enhanced with SCs. The number of SCs per class turns out to be very important, taking too many adds noise to the prompts and lowers the performance. Nevertheless, dynamically choosing the threshold, as described in Sec. 2-Step2c., proves to be a good strategy for adapting the cut-off across classes. Following prior observations regarding the modalities gap between text and image embedding space (Liang et al., 2022; Zhang et al., 2023a), we subtract half of the gap from the embeddings and re-normalize them, ending in marginally lower performance w.r.t. not addressing the gap.

## J.2  IMPACT OF THE LLM CHOSEN FOR FILTERING

To check the importance of the LLM used in the filtering stage we have also employed GPT-3.5 Turbo for the filtering of concepts from the Waterbirds dataset. We provide a qualitative comparison of the responses from Llama-3.1-8B-Instruct and GPT-3.5 Turbo in Tab. 9. For a quantitative comparison we repeat the zero-shot experiment presented in Table 1, with the candidates obtained from GPT-3.5 Turbo. The results are presented in Tab. 10.

Table 9: Ablation. Qualitative comparison of the filtering done by Llama-3.1-8B-Instruct and GPT-3.5 Turbo.

| Original n-gram | Llama-3.1 output | GPT-3.5 Turbo output |
|---|---|---|
| bird flying over a body | flying over a body | flying over a body |
| black bird sitting on top | sitting on top | sitting on top |
| yellow and black bird standing | standing | yellow and black standing |
| bird in a bamboo forest | bamboo forest | in a bamboo forest |
| bird sitting on a boat | boat | sitting on a boat |
| flying over the ocean | flying over the ocean | ocean |
| black bird sitting | sitting | sitting |

Table 10: Ablation. Quantitative comparison of filtering done by Llama-3.1 and GPT-3.5. We use the candidates obtained in both cases for the SC-enhanced zero-shot classification experiment from Tab. 1.

| | Waterbirds (Acc % ↑) | |
|---|---|---|
| Zero-shot | Worst | Avg. |
| **BEE** w/ Llama-3.1 Filtering (in paper) | 50.3 | 86.3 |
| **BEE** w/ GPT-3.5 Filtering | 49.5 | 84.5 |

### J.3  ROLE OF NEGATIVE SCs

Negative SCs of a class hint at the conditions in which said class is less likely to be correctly identified. As such, in a setting where generating and labeling new data points is feasible, the negative SCs indicate what concepts should be depicted near this class in order to mend the SCs learned from the original dataset.

**Binary Classification Tasks (CelebA, Waterbirds, CivilComments)** In binary settings, a positive SC for one class implicitly serves as a negative SC for the other. A positive SC of class 0 favours the prediction of class 0, while implicitly hindering the prediction of class 1. Positive and negative SCs in a task are thus entangled - they share the same underlying pool of concepts. This is also reflected in the symmetry between the positive-SC score of class 0 and the negative-SC score for class 1, for the binary classification setting - $s_{0,i}^{+} = s_{1,i}^{-}$. Proof:

Let $a_i = w_0^{*\top} M(c_i)$ and $b_i = w_1^{*\top} M(c_i)$. We have that $s_{0,i}^{+} = a_i - \min(a_i, b_i)$ and $s_{1,i}^{-} = -b_i - \min(-a_i, -b_i)$.

Using the fact that

$$\min(a_i, b_i) = a_i + b_i - \max(a_i, b_i) = a_i + b_i + \min(-a_i, -b_i)$$

, we obtain the equality:

$$s_{0,i}^{+} = a_i - \min(a_i, b_i) = a_i - a_i - b_i - \min(-a_i, -b_i) = -b_i - \min(-a_i, -b_i) = s_{1,i}^{-}.$$

The SC-enhanced zero-shot classification application from B2T uses the positive SCs for both classes to create more expressive prompts. That is, for a class we use both its positive and negative SCs. We repeated the experiment, using only the positive or only the negative SCs identified by BEE on Waterbirds, leading to significant drops in performance (Tab. 11).

Intuitively, using only the positive SCs for a class amplifies the SCs (we are using prompts of waterbirds in water environments and landbirds in land environments), while using only the negative ones reverses the SCs.

Table 11: Ablation. In the zero-shot SC-augmented prompting setup we use only the positive or only the negative SCs identified for each class.

| Zero-Shot Prompting | Worst Acc % | Avg. Acc % |
|---|---|---|
| with All SCs (in paper) | 50.3 | 86.3 |
| only with Negative SCs | 34.7 | 60.9 |
| only with Positive SCs | 16.7 | 67.9 |

## K  RESULTS OF STATE-OF-THE-ART MODELS

We provide results using the same data and experimental setup used for Tables 2 and 3, for an exhaustive list of ImageNet classifiers, in Tables 12, 13 and 14. Pre-trained models together with their respective weight sets are employed from the `torchvision` package.

Table 12: Accuracy of various convolutional and transformer-based models trained on ImageNet-1k, on the data generated for Tab. 2. As with Fig. 2 and Fig. 1, we note that the performance of these models is significantly affected, even though the correct class is clearly illustrated right in front and center while, and the predicted class is absent from the generated images.

| Model - Weights | Prompt employed (correct class highlighted in bold, SC in red) | | | |
|---|---|---|---|---|
| | a photo of a **peafowl** | firemen and a **peafowl** | a photo of a **Bernese Mountain Dog** | shrimp and pasta near a **Bernese Mountain Dog** |
| alexnet - V1 Krizhevsky (2014) | 100.0 | 4.6 | 96.2 | 23.3 |
| convnext_tiny - V1 Liu et al. (2022) | 100.0 | 77.2 | 94.2 | 69.8 |
| convnext_small - V1 Liu et al. (2022) | 100.0 | 92.9 | 96.3 | 78.3 |
| convnext_base - V1 Liu et al. (2022) | 100.0 | 83.5 | 99.3 | 78.0 |
| convnext_large - V1 Liu et al. (2022) | 100.0 | 88.2 | 99.6 | 81.9 |
| densenet121 - V1 Huang et al. (2016) | 100.0 | 52.3 | 93.1 | 74.8 |
| densenet161 - V1 Huang et al. (2016) | 100.0 | 52.8 | 84.0 | 51.6 |
| densenet201 - V1 Huang et al. (2016) | 100.0 | 49.7 | 86.1 | 80.8 |
| efficientnet_b0 - V1 Tan & Le (2019) | 100.0 | 64.5 | 99.2 | 92.9 |
| efficientnet_b1 - V1 Tan & Le (2019) | 100.0 | 42.6 | 88.1 | 67.1 |
| efficientnet_b1 - V2 Tan & Le (2019) | 100.0 | 84.2 | 99.9 | 69.5 |
| efficientnet_b2 - V1 Tan & Le (2019) | 100.0 | 61.7 | 99.6 | 82.4 |
| efficientnet_b3 - V1 Tan & Le (2019) | 100.0 | 89.1 | 99.1 | 92.8 |
| efficientnet_b4 - V1 Tan & Le (2019) | 100.0 | 94.4 | 99.9 | 72.7 |
| efficientnet_b5 - V1 Tan & Le (2019) | 100.0 | 82.9 | 99.4 | 86.3 |
| efficientnet_b6 - V1 Tan & Le (2019) | 100.0 | 91.2 | 100.0 | 95.6 |
| efficientnet_b7 - V1 Tan & Le (2019) | 100.0 | 88.3 | 99.9 | 90.3 |
| efficientnet_v2_s - V1 Tan & Le (2021) | 100.0 | 98.3 | 99.2 | 88.1 |
| efficientnet_v2_m - V1 Tan & Le (2021) | 100.0 | 95.1 | 99.8 | 94.3 |
| efficientnet_v2_l - V1 Tan & Le (2021) | 100.0 | 96.0 | 99.1 | 83.6 |
| googlenet - V1 Szegedy et al. (2015) | 100.0 | 45.4 | 95.2 | 57.0 |
| inception_v3 - V1 Szegedy et al. (2016) | 100.0 | 82.2 | 98.9 | 80.8 |
| maxvit_t - V1 Tu et al. (2022) | 100.0 | 91.7 | 99.7 | 85.3 |
| mnasnet0_5 - V1 Tan et al. (2019) | 100.0 | 23.3 | 96.7 | 60.8 |
| mnasnet0_75 - V1 Tan et al. (2019) | 100.0 | 36.9 | 98.4 | 71.1 |
| mnasnet1_0 - V1 Tan et al. (2019) | 100.0 | 32.0 | 89.4 | 74.9 |
| mnasnet1_3 - V1 Tan et al. (2019) | 100.0 | 63.9 | 91.7 | 75.6 |
| mobilenet_v2 - V1 Sandler et al. (2018) | 100.0 | 26.2 | 84.0 | 45.0 |
| mobilenet_v2 - V2 Sandler et al. (2018) | 100.0 | 54.1 | 98.6 | 62.8 |
| mobilenet_v3_small - V1 Howard et al. (2019) | 100.0 | 16.6 | 91.0 | 30.3 |
| mobilenet_v3_large - V1 Howard et al. (2019) | 100.0 | 25.4 | 94.2 | 23.1 |
| mobilenet_v3_large - V2 Howard et al. (2019) | 100.0 | 56.0 | 96.6 | 73.7 |

Table 13: Accuracy of various convolutional and transformer-based models trained on ImageNet-1k, on the data generated for Tab. 2. As with Fig. 1, we note that the performance of these models is significantly affected, even though the correct class is clearly illustrated right in front and center while, and the predicted class is absent from the generated images.

| Model - Weights | Prompt employed (correct class highlighted in bold, SC in red) | | | |
|---|---|---|---|---|
| | a photo of a **peafowl** | firemen and a **peafowl** | a photo of a **Bernese Mountain Dog** | shrimp and pasta near a **Bernese Mountain Dog** |
| regnet_y_400mf - V1 Radosavovic et al. (2020) | 100.0 | 35.6 | 74.7 | 34.7 |
| regnet_y_400mf - V2 Radosavovic et al. (2020) | 100.0 | 72.1 | 97.7 | 87.9 |
| regnet_y_800mf - V1 Radosavovic et al. (2020) | 100.0 | 22.7 | 94.8 | 54.4 |
| regnet_y_800mf - V2 Radosavovic et al. (2020) | 100.0 | 81.7 | 98.9 | 88.0 |
| regnet_y_1_6gf - V1 Radosavovic et al. (2020) | 100.0 | 47.0 | 96.4 | 71.7 |
| regnet_y_1_6gf - V2 Radosavovic et al. (2020) | 100.0 | 88.2 | 96.4 | 74.1 |
| regnet_y_3_2gf - V1 Radosavovic et al. (2020) | 100.0 | 33.5 | 99.4 | 90.9 |
| regnet_y_3_2gf - V2 Radosavovic et al. (2020) | 100.0 | 94.6 | 98.4 | 75.4 |
| regnet_y_8gf - V1 Radosavovic et al. (2020) | 100.0 | 58.3 | 71.3 | 54.8 |
| regnet_y_8gf - V2 Radosavovic et al. (2020) | 100.0 | 98.2 | 96.8 | 67.5 |
| regnet_y_16gf - V1 Radosavovic et al. (2020) | 100.0 | 86.7 | 97.8 | 49.8 |
| regnet_y_16gf - V2 Radosavovic et al. (2020) | 100.0 | 98.7 | 91.8 | 83.5 |
| regnet_y_16gf - SWAG_E2E_V1 Radosavovic et al. (2020) | 100.0 | 99.6 | 97.7 | 78.2 |
| regnet_y_16gf - SWAG_LINEAR_V1 Radosavovic et al. (2020) | 100.0 | 78.3 | 100.0 | 90.7 |
| regnet_y_32gf - V1 Radosavovic et al. (2020) | 100.0 | 84.5 | 99.1 | 82.3 |
| regnet_y_32gf - V2 Radosavovic et al. (2020) | 100.0 | 98.0 | 99.5 | 79.6 |
| regnet_y_32gf - SWAG_E2E_V1 Radosavovic et al. (2020) | 100.0 | 99.6 | 93.5 | 71.5 |
| regnet_y_32gf - SWAG_LINEAR_V1 Radosavovic et al. (2020) | 100.0 | 97.2 | 100.0 | 85.6 |
| regnet_y_128gf - SWAG_E2E_V1 Radosavovic et al. (2020) | 100.0 | 99.6 | 54.4 | 67.8 |
| regnet_y_128gf - SWAG_LINEAR_V1 Radosavovic et al. (2020) | 100.0 | 90.8 | 99.9 | 96.9 |
| regnet_x_400mf - V1 Radosavovic et al. (2020) | 100.0 | 37.4 | 82.3 | 29.0 |
| regnet_x_400mf - V2 Radosavovic et al. (2020) | 100.0 | 50.7 | 99.5 | 81.7 |
| regnet_x_800mf - V1 Radosavovic et al. (2020) | 100.0 | 25.4 | 77.0 | 52.6 |
| regnet_x_800mf - V2 Radosavovic et al. (2020) | 100.0 | 75.5 | 97.1 | 71.4 |
| regnet_x_1_6gf - V1 Radosavovic et al. (2020) | 100.0 | 38.2 | 76.9 | 66.7 |
| regnet_x_1_6gf - V2 Radosavovic et al. (2020) | 100.0 | 82.2 | 99.2 | 88.1 |
| regnet_x_3_2gf - V1 Radosavovic et al. (2020) | 100.0 | 45.1 | 62.1 | 69.4 |
| regnet_x_3_2gf - V2 Radosavovic et al. (2020) | 100.0 | 83.4 | 99.6 | 88.3 |
| regnet_x_8gf - V1 Radosavovic et al. (2020) | 100.0 | 41.8 | 98.8 | 89.0 |
| regnet_x_8gf - V2 Radosavovic et al. (2020) | 100.0 | 93.5 | 99.2 | 81.6 |
| regnet_x_16gf - V1 Radosavovic et al. (2020) | 100.0 | 48.7 | 86.6 | 54.5 |
| regnet_x_16gf - V2 Radosavovic et al. (2020) | 100.0 | 93.4 | 97.9 | 88.1 |
| regnet_x_32gf - V1 Radosavovic et al. (2020) | 100.0 | 66.1 | 85.9 | 46.0 |
| regnet_x_32gf - V2 Radosavovic et al. (2020) | 100.0 | 97.4 | 99.6 | 83.7 |
| resnet18 - V1 He et al. (2016) | 100.0 | 36.8 | 84.3 | 41.9 |
| resnet34 - V1 He et al. (2016) | 100.0 | 32.9 | 54.1 | 26.1 |
| resnet50 - V1 He et al. (2016) | 100.0 | 30.1 | 73.9 | 54.5 |
| resnet50 - V2 He et al. (2016) | 100.0 | 80.1 | 99.7 | 88.4 |
| resnet101 - V1 He et al. (2016) | 100.0 | 60.4 | 92.3 | 84.4 |
| resnet101 - V2 He et al. (2016) | 100.0 | 93.4 | 98.8 | 87.4 |
| resnet152 - V1 He et al. (2016) | 100.0 | 66.1 | 98.4 | 78.2 |
| resnet152 - V2 He et al. (2016) | 100.0 | 93.1 | 98.5 | 92.5 |
| resnext50_32x4d - V1 Xie et al. (2017) | 100.0 | 45.5 | 92.6 | 74.8 |
| resnext50_32x4d - V2 Xie et al. (2017) | 100.0 | 80.3 | 98.5 | 88.0 |
| resnext101_32x8d - V1 Xie et al. (2017) | 100.0 | 66.6 | 84.7 | 61.2 |
| resnext101_32x8d - V2 Xie et al. (2017) | 100.0 | 90.3 | 99.4 | 85.2 |
| resnext101_64x4d - V1 Xie et al. (2017) | 100.0 | 77.9 | 97.7 | 74.2 |

Table 14: Accuracy of various convolutional and transformer-based models trained on ImageNet-1k, on the data generated for Tab. 2. As with Fig. 2 and Fig. 1, we note that the performance of these models is significantly affected, even though the correct class is clearly illustrated right in front and center while, and the predicted class is absent from the generated images.

| Model - Weights | Prompt employed (correct class highlighted in bold, SC in red) | | | |
| --- | --- | --- | --- | --- |
| | a photo of a **peafowl** | firemen and a **peafowl** | a photo of a **Bernese Mountain Dog** | shrimp and pasta near a **Bernese Mountain Dog** |
| shufflenet_v2_x0_5 - V1 Ma et al. (2018) | 100.0 | 23.8 | 36.9 | 21.0 |
| shufflenet_v2_x1_0 - V1 Ma et al. (2018) | 99.8 | 30.4 | 72.1 | 55.9 |
| shufflenet_v2_x1_5 - V1 Ma et al. (2018) | 100.0 | 41.2 | 97.9 | 52.5 |
| shufflenet_v2_x2_0 - V1 Ma et al. (2018) | 100.0 | 61.4 | 99.3 | 64.5 |
| squeezenet1_0 - V1 Iandola et al. (2016) | 100.0 | 11.4 | 95.9 | 28.7 |
| squeezenet1_1 - V1 Iandola et al. (2016) | 100.0 | 13.8 | 91.2 | 46.1 |
| swin_t - V1 Liu et al. (2021b) | 100.0 | 72.7 | 96.9 | 81.6 |
| swin_s - V1 Liu et al. (2021b) | 100.0 | 74.3 | 99.3 | 81.4 |
| swin_b - V1 Liu et al. (2021b) | 100.0 | 81.5 | 95.2 | 72.6 |
| swin_v2_t - V1 Liu et al. (2021b) | 100.0 | 76.2 | 88.7 | 73.6 |
| swin_v2_s - V1 Liu et al. (2021b) | 100.0 | 85.7 | 90.7 | 74.4 |
| swin_v2_b - V1 Liu et al. (2021b) | 100.0 | 73.0 | 96.0 | 85.2 |
| vgg11 - V1 Simonyan & Zisserman (2015) | 100.0 | 9.9 | 96.6 | 60.7 |
| vgg11_bn - V1 Simonyan & Zisserman (2015) | 100.0 | 15.9 | 86.7 | 54.8 |
| vgg13 - V1 Simonyan & Zisserman (2015) | 100.0 | 5.9 | 97.7 | 68.1 |
| vgg13_bn - V1 Simonyan & Zisserman (2015) | 100.0 | 15.5 | 87.0 | 13.0 |
| vgg16 - V1 Simonyan & Zisserman (2015) | 100.0 | 12.2 | 93.7 | 66.0 |
| vgg16_bn - V1 Simonyan & Zisserman (2015) | 100.0 | 22.6 | 96.1 | 70.7 |
| vgg19 - V1 Simonyan & Zisserman (2015) | 100.0 | 26.6 | 98.5 | 40.8 |
| vgg19_bn - V1 Simonyan & Zisserman (2015) | 100.0 | 35.9 | 83.1 | 46.2 |
| vit_b_16 - V1 Dosovitskiy et al. (2021) | 100.0 | 85.4 | 97.1 | 75.2 |
| vit_b_16 - SWAG_E2E_V1 Dosovitskiy et al. (2021) | 100.0 | 88.9 | 95.8 | 59.1 |
| vit_b_16 - SWAG_LINEAR_V1 Dosovitskiy et al. (2021) | 100.0 | 79.2 | 99.9 | 84.6 |
| vit_b_32 - V1 Dosovitskiy et al. (2021) | 100.0 | 56.1 | 96.0 | 86.0 |
| vit_l_16 - V1 Dosovitskiy et al. (2021) | 100.0 | 55.9 | 95.3 | 76.0 |
| vit_l_16 - SWAG_E2E_V1 Dosovitskiy et al. (2021) | 100.0 | 92.1 | 100.0 | 93.3 |
| vit_l_16 - SWAG_LINEAR_V1 Dosovitskiy et al. (2021) | 100.0 | 97.4 | 100.0 | 72.5 |
| vit_l_32 - V1 Dosovitskiy et al. (2021) | 100.0 | 54.0 | 96.9 | 82.9 |
| vit_h_14 - SWAG_E2E_V1 Dosovitskiy et al. (2021) | 100.0 | 99.4 | 98.5 | 86.4 |
| vit_h_14 - SWAG_LINEAR_V1 Dosovitskiy et al. (2021) | 100.0 | 99.7 | 100.0 | 96.1 |
| wide_resnet50_2 - V1 Zagoruyko & Komodakis (2016) | 100.0 | 60.6 | 95.7 | 63.9 |
| wide_resnet50_2 - V2 Zagoruyko & Komodakis (2016) | 100.0 | 83.9 | 99.5 | 85.4 |
| wide_resnet101_2 - V1 Zagoruyko & Komodakis (2016) | 100.0 | 69.3 | 84.1 | 72.8 |
| wide_resnet101_2 - V2 Zagoruyko & Komodakis (2016) | 100.0 | 91.0 | 98.8 | 84.9 |

Table 15: Top Waterbirds class-neutral concepts for "landbird".

| Landbird | Score |
|---|---|
| forest floor | 0.055562317 |
| forest next to a tree | 0.053587496 |
| bamboo forest floor | 0.05134508 |
| forest of bamboo | 0.04781133 |
| forest | 0.047080815 |
| snowy forest | 0.044688106 |
| ground | 0.043406844 |
| field | 0.043168187 |
| log | 0.043052554 |
| standing on a forest floor | 0.041143 |
| grass covered | 0.040526748 |
| tree branch in a forest | 0.039670765 |
| forest with trees | 0.03949821 |
| tree in a forest | 0.039123535 |
| bamboo forest | 0.03876221 |
| front of bamboo | 0.037381053 |
| mountain | 0.036155403 |
| forest of trees | 0.03600967 |
| flying through a forest | 0.035692394 |
| platform | 0.03565806 |
| standing in a forest | 0.034528017 |
| hill | 0.03341371 |

Table 16: Top Waterbirds class-neutral concepts for "waterbird".

| Waterbird | Score |
|---|---|
| swimming in the water | 0.11482495 |
| water lily | 0.10905403 |
| boat in the water | 0.1066975 |
| floating in the water | 0.106155455 |
| water | 0.106134474 |
| flying over the water | 0.10561061 |
| standing in the water | 0.10444009 |
| sitting in the water | 0.103776515 |
| body of water | 0.09977633 |
| water in front | 0.0902465 |
| standing in water | 0.086544394 |
| water and one | 0.08424729 |
| swimming | 0.07818574 |
| standing on a lake | 0.06565446 |
| flying over the ocean | 0.06509364 |
| flying over a pond | 0.06463468 |
| boats | 0.061231434 |
| lifeguard | 0.06122452 |
| flying over a lake | 0.060126305 |
| boat | 0.0571931 |
| pond | 0.053261578 |

Table 17: Top CelebA class-neutral concepts for "non-blonde".

| Non-Blonde | Score |
|---|---|
| hat on and a blue | 0.13952243 |
| hat on and a man | 0.13853341 |
| man in the hat | 0.13803285 |
| man who made | 0.13307464 |
| man behind | 0.13247031 |
| man with the hat | 0.13186401 |
| man is getting | 0.12861347 |
| actor | 0.12726557 |
| dark | 0.12713176 |
| man in a blue | 0.1269682 |
| person | 0.12541258 |
| man in the blue | 0.124844134 |
| man is not a man | 0.12308431 |
| man | 0.12290484 |
| large | 0.1228559 |
| shirt on in a dark | 0.12170941 |
| hat | 0.121646166 |
| close | 0.12146461 |
| man with the blue | 0.12136656 |
| man face | 0.12130207 |

Table 18: Top CivilComments class-neutral concepts for "non-offensive".

| Non-offensive | Score |
| --- | --- |
| allowing | 0.07341421 |
| work | 0.06982881 |
| made | 0.069063246 |
| talk | 0.06858361 |
| none are needed | 0.067236125 |
| check | 0.06664443 |
| helping keep the present | 0.06531584 |
| policy | 0.06339955 |
| campaign | 0.06333798 |
| involved in the first place | 0.063222766 |
| Cottage | 0.063149124 |
| IDEA | 0.06310266 |
| stories | 0.0625782 |
| job | 0.06236595 |
| allowed | 0.062137783 |
| latest news about the origin | 0.062061936 |
| giving others who have experienced | 0.061925888 |
| proposed | 0.061897278 |
| one purpose | 0.06122935 |
| starting | 0.061154723 |
| small | 0.061071455 |
| question | 0.060854554 |
| practice | 0.060740173 |
| raised | 0.060681045 |
| entering | 0.060585797 |
| registered | 0.060475767 |
| beliefs | 0.060165346 |
| accept that they are promoting | 0.060070753 |
| Security | 0.059328556 |
| new | 0.059324086 |
| subject | 0.058983028 |
| close | 0.058632135 |
| views | 0.058573127 |
| Hold | 0.058341324 |
| reality for a change | 0.058261245 |
| built at that parish | 0.057885766 |
| rest | 0.057804525 |
| historic | 0.057656527 |
| concept | 0.057422698 |
| people | 0.057151675 |
| passage seems to in reflection | 0.05699992 |
| attempt | 0.056797385 |

Table 19: Top CivilComments class-neutral concepts for "offensive".

| "Offensive" – Top Concepts | Score |
| --- | --- |
| hypocrisy | 0.046756804 |
| troll | 0.035944045 |
| silly | 0.029536605 |
| hate | 0.013704538 |
| silly how do you study | 0.00325954 |
| spite | 0.002645433 |
| kid you have the absolute | 0.001619577 |

Table 20: Top ImageNet class-neutral concepts for "Crossword".

| "Crossword" – Top Concepts | Score |
| --- | --- |
| reading a newspaper | 0.30469692 |
| man reading a newspaper | 0.29045385 |
| crochet squares | 0.28316277 |
| sitting on a newspaper | 0.27749887 |
| newspaper sitting | 0.27106437 |
| crochet squares in a square | 0.2708223 |
| newspaper that has the words | 0.26934764 |
| holding a newspaper | 0.26375395 |
| newspaper while sitting | 0.26288068 |
| newspaper laying | 0.25538272 |
| square with a few crochet | 0.2461972 |
| square with a crochet | 0.24225119 |
| square of crochet squares | 0.23986068 |
| checkerboard | 0.23797607 |
| crochet blanket in a square | 0.2364017 |
| square of square crochet | 0.23590976 |
| on a newspaper | 0.2357213 |
| crochet square with a crochet | 0.23569846 |
| crochet blanket with a crochet | 0.23438567 |
| newspaper | 0.23315597 |
| crochet with a square | 0.23304509 |
| crochet square | 0.23259673 |
| newspaper sitting on | 0.23098715 |
| square of crochet yarn | 0.2291883 |
| square crochet | 0.22903368 |
| crochet blanket with a square | 0.22888878 |
| crochet square sitting | 0.22860557 |
| crochet in a square | 0.22856355 |
| square with a single crochet | 0.22752959 |
| free crochet | 0.22748157 |
| newspaper with | 0.22672665 |
| is on a newspaper | 0.2257084 |
| crochet blanket | 0.22547376 |
| checkered blanket | 0.22514643 |
| square of crochet | 0.22324148 |

Table 21: Top ImageNet class-neutral concepts for "Guacamole".

| "Guacamole" – Top Concepts | Score |
| --- | --- |
| tomatoes and avocado | 0.33948907 |
| avocado | 0.3125001 |
| nachos | 0.30761188 |
| ham and parsley | 0.2960047 |
| with avocado | 0.2952027 |
| colorful mexican | 0.28313732 |
| bacon and parsley | 0.27773544 |
| of avocado | 0.2754345 |
| side of salsa | 0.27470407 |
| peridot | 0.2734925 |
| salsa | 0.27124816 |
| nachos with | 0.27092364 |
| lime body | 0.27088284 |
| cheese and parsley | 0.26951405 |
| pile of limes | 0.26944226 |
| peas and bacon | 0.2671204 |
| limes and limes | 0.26682717 |
| lime cut | 0.2662533 |
| nachos with and | 0.26580203 |
| pasta with peas | 0.2626204 |
| tacos | 0.2619322 |
| pasta with ham and parsley | 0.26183394 |
| tomatoes and cilantro | 0.26161948 |

Table 23: Top ImageNet class-neutral concepts for "Ballpoint Pen".

Table 22: Top ImageNet class-neutral concepts for "Bald Eagle".

| "Bald Eagle" – Top Concepts | Score |
| --- | --- |
| osprey flying | 0.2888234 |
| osprey | 0.27377927 |
| row of american flags | 0.26513425 |
| group of american flags flying | 0.25681192 |
| american flag and american flag | 0.23945728 |
| emu standing | 0.23752406 |
| yellowstone national | 0.23740456 |

| "Ballpoint Pen" – Top Concepts | Score |
| --- | --- |
| wearing a pilot | 0.36252645 |
| markers | 0.36050144 |
| sharpie | 0.34563553 |
| stylus | 0.34399948 |
| pair of eyeglasses | 0.32426757 |
| notepad | 0.3228797 |
| calligraphy | 0.3222967 |
| paint and markers | 0.32198787 |
| close up of a needle | 0.32131955 |
| on a straw | 0.3209229 |
| and markers | 0.32090995 |
| eyeglasses | 0.31017447 |
| dots | 0.30764964 |
| crayons | 0.30598855 |

Table 24: Top ImageNet class-neutral concepts for "Coffeemaker".

Table 25: Top ImageNet class-neutral concepts for "Doormat".

| "Coffeemaker" – Top Concepts | Score |
|---|---|
| kettle sitting on | 0.43714887 |
| with a kettle | 0.4207235 |
| thermos | 0.40833473 |
| kettle | 0.40526068 |
| kettle sitting | 0.3881535 |
| stovetop maker | 0.37747166 |
| kettle kettle | 0.37671012 |
| kettle kettle kettle kettle | 0.37649006 |
| vases and vases | 0.37567452 |
| kettle kettle kettle | 0.37547356 |
| flask | 0.37485123 |
| kettle kettle kettle kettle kettle | 0.37305972 |
| large canister | 0.37186915 |
| flasks | 0.36940324 |
| decorative vases sitting | 0.3683877 |
| kitchen aid | 0.36793774 |
| milkshakes | 0.36605325 |
| large pottery | 0.36506 |
| set of kitchen | 0.3650242 |
| cookbook | 0.36469316 |
| vases sitting | 0.3643943 |

| "Doormat" – Top Concepts | Score |
|---|---|
| brick sidewalk | 0.37059835 |
| laying on gravel | 0.34978455 |
| laying on a carpeted | 0.34279323 |
| crochet blanket | 0.3421431 |
| sitting on a carpeted | 0.34104648 |
| laying on a step | 0.3400078 |
| crochet blanket in a square | 0.33642814 |
| brick walkway | 0.3319164 |
| carpeted floor | 0.33067068 |
| on a brick sidewalk | 0.32878387 |
| crochet blanket with a crochet | 0.32599914 |
| floor with a welcome | 0.32548892 |
| square of crochet yarn | 0.32428077 |
| crochet blanket made | 0.32336423 |
| carpeted staircase | 0.3227385 |
| dot blanket | 0.3213501 |
| mosaic floor | 0.32125634 |
| crocheted blanket | 0.31886423 |
| on a blanket | 0.3182252 |
| crochet blanket with a square | 0.31753486 |
| square of crochet squares | 0.3169018 |
| crochet squares | 0.3152317 |
| standing in a doorway | 0.3126963 |

Table 26: Top ImageNet class-neutral concepts for "Eraser".

Table 27: Top ImageNet class-neutral concepts for "American Lobster".

| "Eraser" – Top Concepts | Score |
|---|---|
| crayons | 0.38733196 |
| chalk | 0.37349075 |
| graphite | 0.36778685 |
| crayon | 0.36178917 |
| charcoal | 0.351962 |
| lip balm lip | 0.35033816 |
| band aid cookie | 0.34825876 |
| lip balm | 0.3409983 |
| nose sticking | 0.34016216 |
| sticking | 0.33971623 |
| markers | 0.33940658 |
| matchbox | 0.33480892 |
| wand | 0.33415005 |
| stylus | 0.3318595 |
| band aid card | 0.33174193 |
| toothbrush | 0.33009088 |
| office supplies | 0.32956824 |
| band aid flexible | 0.32946587 |

| "American Lobster" – Top Concepts | Score |
|---|---|
| pasta with shrimp | 0.3148532 |
| adirondack sitting | 0.31178916 |
| large shrimp | 0.3085378 |
| shrimp and pasta | 0.30821544 |
| shrimp cooking | 0.3032636 |
| large cast cooking | 0.30282205 |
| pasta with shrimp and cheese | 0.2984723 |
| adirondack | 0.29169592 |
| and mussels | 0.2904386 |
| legs and other seafood | 0.28870153 |
| close up of a shrimp | 0.2871693 |
| of pasta with shrimp | 0.2823377 |
| seafood | 0.2808096 |
| mussels | 0.28018713 |
| cast cooking | 0.27951774 |
| shrimp and cheese | 0.27841187 |
| shrimp | 0.2726224 |

Table 28: Top ImageNet class-neutral concepts for "Fire Truck".

| "Fire Truck" – Top Concepts | Score |
|---|---|
| firefighter spraying | 0.34155905 |
| group of firefighters | 0.3347583 |
| firefighters | 0.33051446 |
| firefighter | 0.31450543 |
| firefighter wearing | 0.2843979 |
| hydrant spraying | 0.26793447 |
| firefighter wearing a | 0.26590723 |
| firefighter cuts | 0.2613345 |
| farmall parked | 0.2591076 |
| dashboard with flames | 0.25825307 |
| flames painted | 0.2540929 |

