# OpenReview forum: "Bridging Explainability and Embeddings: BEE Aware of Spuriousness"
_ICLR.cc/2026/Conference — ICLR 2026 Poster_

### Official Review · Reviewer_Qgvj · 2025-10-27

**Soundness:** 3
**Presentation:** 2
**Contribution:** 3
**Rating:** 6
**Confidence:** 3

**Summary:**

This paper proposes BEE, a weight-space diagnostic to identify class-specifc spurious correlations learned during fine-tuning. Specifically, it can be divivided into two steps: (1) Initialize linear probe with zero-shot class embeddings from a foundation model and finetune the probe; (2) Rank class-neutral concepts by their similartiy in terms of class weights and thresholding to select SCs. The method is evauate across different modalities including vision (e.g., ImageNet) and language (e.g., CivilComments) with embeddings from different foundation models. It shows strong generalization ability across different tasks with spurious correlation settings.

**Strengths:**

1. The author proposes a novel method with clear motivation. Rather than error- or data-driven SC doscovery, them operate the measurement in weight space. This makes spurious concept can be dicsovered from the geometry of embedding space, which is simple and aligned with human intuition.

2. The experiments are evaluated on broad scope with multi-modal validation, demonstrating strong generalization. The pipeline is also simple and easy-to-understand, which can improve its practical usage.

**Weaknesses:**

1. The concept extraction, LLM filtering and dynamic thresholding the spurious concepts are heuristic and heavily relied on the other foundation model which can potentially also suffer from spurious correlation.

2. The gemoetric view is novel, but the paper lacks theoretical analysis about when weight drift towards concept embeddings necessarily indicates spuriousness versus legitimate subclass structure.

3. The defnition of class-neutral is ambiguous. The filtering relies on LLM heuristics. It's possible to assign the attributes relevant to class context (but not spurious) incorrectly as spurious concept. It will be better to discuss it with precison/recall of the class-neutral label.

**Questions:**

1. Could you provide the details of the computation overhead for the spurious correlation detection?

2. Is there any broader impact of detecting such spurious correlation apart from classification? Will it affect downstream tasks?

---

> ### Author Response · Authors · 2025-11-20
>
> Dear Reviewer Qgvj,
>
> We sincerely thank you for the thoughtful evaluation and constructive feedback. We are grateful for your recognition of our novel weight-space perspective and the broad experimental validation across modalities. Your insightful observations regarding the precision of class-neutral concept filtering and the reliance on foundation models are well-taken and insightful. We hope we can address some of your concerns as follows:
>
> ---
>
> > W1. The concept extraction, LLM filtering and dynamic thresholding the spurious concepts are heuristic and heavily relied on the other foundation model which can potentially also suffer from spurious correlation.
>
> We thank the reviewer for this valuable observation. We agree that this aspect could be problematic; however, these components are used solely for candidate generation and filtering and therefore have limited influence on the core diagnostic process. Potential biases in these auxiliary steps could, in principle, lead to either over-filtering or under-filtering:
>
> - Over-filtering (too few candidates) could exclude legitimate spurious cues, resulting in fewer or overly trivial SCs being identified. In practice, this was not observed, as BEE consistently revealed new and meaningful spurious correlations, including ones not captured by previous methods (see ImageNet results).
> - Under-filtering (too many candidates) could include concepts genuinely related to the class, mistakenly labeling them as spurious. However, our experiments did not show this effect either, indicating that if it occurs, it does not happen at scale. As detailed in the responses on subclass structure (W2) and class-neutral definition (W3), BEE effectively distinguishes between class-related and spurious concepts. In all cases we tested, introducing the identified SCs consistently reduced model robustness, confirming their spurious nature (e.g., Tables 2 and 3).
>
> Overall, while these heuristic steps facilitate scalability, the core diagnostic signal in BEE stems from the geometric alignment between the fine-tuned model’s class weights and concept embeddings, which is independent of potential auxiliary model biases.
>
> > W2. The geometric view is novel, but the paper lacks theoretical analysis about when weight drift towards concept embeddings necessarily indicates spuriousness versus legitimate subclass structure.
>
> > W3. The defnition of class-neutral is ambiguous. The filtering relies on LLM heuristics. It's possible to assign the attributes relevant to class context (but not spurious) incorrectly as spurious concept. It will be better to discuss it with precison/recall of the class-neutral label.
>
> We thank the reviewer for this insightful comment. Our definition of class-neutral concepts relies on hierarchical WordNet filtering and LLM-based checks, which are specifically designed to filter out the subclasses structure from the spurious candidate pool. By removing concepts that are semantically related or hierarchically linked to the target class, this filtering aims to exclude attributes that are truly part of the class definition, rather than spurious context.
>
> Nevertheless, we acknowledge that these mechanisms are heuristic, and some legitimate subclass attributes may occasionally be flagged as spurious. However, our empirical results suggest that such cases are rare and do not dominate the identified set. Across multiple datasets and modalities, the concepts detected by BEE, when used into the inputs, consistently lead toward confirming their spuriousness (Tables 2 and 3).
>
> In summary, the combination of semantic hierarchy checks and LLM validation provides strong evidence that BEE primarily surfaces genuinely spurious rather than legitimate subclass correlations.

---

> ### Author Response · Authors · 2025-11-20
>
> > Q1. Could you provide the details of the computation overhead for the spurious correlation detection?
>
> Preprocessing is fully automated and critical for SC quality. For ImageNet-1k, the most expensive dataset to preprocess, it takes ~12 hours on an RTX 4090, incurred only once per dataset. Afterward, models can be evaluated efficiently, with linear probing taking ~2 minutes per model.
>
> > Q2. Is there any broader impact of detecting such spurious correlation apart from classification? Will it affect downstream tasks?
>
> This is a very good question. The spurious correlations we detect often reflect co-occurrences inherently present in the dataset. For example, in ImageNet, firefighters frequently appear together with fire trucks. Such co-occurrences are task-agnostic and remain in the data regardless of the specific downstream objective. This means that any model trained on the same dataset, whether for classification, VQA, or another task, can implicitly learn these associations. For instance, a VQA model trained on such data might learn to answer “yes” when asked whether a fire truck is present whenever it sees a firefighter, even if none is shown.
>
> Therefore, what BEE reveals are root causes of shortcut learning that can propagate across tasks built on the same dataset. Although their exact effect depends on the task, these patterns represent general dataset-level dependencies that are valuable to detect and understand beyond classification. While we did not explicitly test the identified spurious correlations on other downstream tasks, these dataset-level co-occurrences could transfer and influence models trained on the same data.

---

### Official Review · Reviewer_Auj2 · 2025-10-30

**Soundness:** 3
**Presentation:** 3
**Contribution:** 2
**Rating:** 6
**Confidence:** 3

**Summary:**

This paper introduces BEE (Bridging Explainability and Embeddings), a novel framework for detecting spurious correlations in fine-tuned foundation models without requiring counterexamples or group annotations.

The method uses a transparent linear probing setup, identifies class-neutral concepts in the embedding space, and ranks them based on their alignment with classifier weights.

Experiments cover a wide range of modalities (vision, language, healthcare) and embedding models (CLIP, BLIP2, mGTE, etc.).

**Strengths:**

- a conceptually elegant and model-agnostic approach to uncover spurious correlations from classifier weights, which is both simple and broadly applicable.

- tightly aligned with foundation models: BEE leverages their shared embedding spaces to analyze both classifier weights and textual concepts. This makes BEE natively compatible with large pre-trained models, including CLIP, BLIP2, mGTE, and others.

- strong empirical results on diverse tasks and datasets.

- the methods provides explicit concept-level explanations for model decisions, thus good interpretability.

**Weaknesses:**

- BEE operates entirely within the embedding space of large foundation models such as CLIP or mGTE. If the embedding model itself has already encoded biased or spurious associations, BEE may merely surface these existing biases, rather than revealing new or independent shortcuts learned during downstream training. This raises the question of whether BEE is diagnosing the fine-tuning process or simply interpreting the biases already present in the frozen embeddings.

- BEE relies on linear probing to identify correlations between classifier weights and concept embeddings. As such, it is inherently limited to capturing linear or near-linear relationships. What will happen if BEE is applied on more complex spurious patterns?

- BEE does not offer any mechanism for diagnosing or correcting the biases that may exist in the foundation model’s embedding space. This is a critical limitation, as the root causes of many spurious correlations may lie within the embedding model itself.

**Questions:**

NA

---

> ### Author Response · Authors · 2025-11-20
>
> Dear Reviewer Auj2,
>
> We sincerely thank you for the thoughtful and constructive feedback. We greatly appreciate the recognition of BEE's conceptual elegance, broad applicability across modalities, and strong empirical results. The concerns raised regarding the potential conflation of embedding-space biases with downstream spurious correlations and the limitations of linear probing for complex patterns are well-taken and insightful. We hope we can address some of your concerns as follows:
>
> ---
>
> > W1. BEE operates entirely within the embedding space of large foundation models such as CLIP or mGTE. If the embedding model itself has already encoded biased or spurious associations, BEE may merely surface these existing biases, rather than revealing new or independent shortcuts learned during downstream training. This raises the question of whether BEE is diagnosing the fine-tuning process or simply interpreting the biases already present in the frozen embeddings.
>
> We thank the reviewer for this insightful observation. Through our experiments, we found empirical evidence suggesting that these biases are more strongly linked to the downstream dataset than to the foundation model itself, as described below. However, we acknowledge that formally disentangling these two sources remains an open and challenging research problem. In the end, BEE does not aim to separate the sources of bias, but rather to identify the spurious correlations that the fine-tuned model actually relies on during prediction.
>
> **Empirical evidence.** On datasets with known spurious correlations (e.g., CelebA, Waterbirds), BEE successfully recovered the expected shortcuts, indicating that it primarily captures dataset-level effects. On ImageNet, we further observed that the same BEE-identified concepts caused performance degradation across multiple architectures trained independently of CLIP (e.g., EfficientNet, ResNet, ViT; see Table 3). This consistency suggests that the surfaced correlations are largely learned during downstream training, rather than inherited from the frozen embeddings.
>
> **Practical takeaway.** While we do not explicitly disentangle these bias sources, our findings point to a stronger dataset influence. Inspired by the reviewer’s suggestion, a promising future direction would be to analyze the intersection of spurious correlations identified using multiple foundation models trained on diverse data, which could provide a clearer view of which biases originate from the dataset versus the embedding space.
>
> > W2. BEE relies on linear probing to identify correlations between classifier weights and concept embeddings. As such, it is inherently limited to capturing linear or near-linear relationships. What will happen if BEE is applied on more complex spurious patterns?
>
> We thank the reviewer for raising this important point. We agree that BEE relies on linear probing, which may at first appear limited to linear or near-linear relationships. However, this aligns with findings showing that such relationships are often sufficient for diagnosing spuriousness in the final representation layer. Prior work (e.g., Kirichenko et al., Last Layer Re-Training is Sufficient for Robustness to Spurious Correlations, ICLR 2023) shows that spurious and core features tend to be linearly separable in the final representation layer, supporting our methodological choice. Additionally, recent findings using Sparse Autoencoders (e.g., Robert Huben et al., Sparse Autoencoders Find Highly Interpretable Features in Language Models, ICLR 2024) demonstrate that many meaningful and disentangled features can be isolated within a single linear layer.
>
> Based on these findings, we consider that linear probing provides an effective balance between expressiveness and interpretability, making it well suited for diagnosing spurious correlations.

---

> ### Author Response · Authors · 2025-11-20
>
> > W3. BEE does not offer any mechanism for diagnosing or correcting the biases that may exist in the foundation model’s embedding space. This is a critical limitation, as the root causes of many spurious correlations may lie within the embedding model itself.
>
> **Biases in the FM’s embedding space:** As discussed in W1, disentangling the biases originating from the foundation model’s embedding space and those arising from the downstream dataset remains a challenging and open research problem that BEE does not aim to solve. Our goal is not to claim a perfect separation between these two sources of bias, but rather to identify the spurious correlations that the final, fine-tuned model actually relies on during prediction. Moreover, as detailed in W2, such biases can ultimately be addressed simultaneously within a linear layer.
>
> **Correcting the biases (mitigation):** We acknowledge that BEE itself is not a mitigation method - its focus is diagnostic and explanatory. However, by naming and localizing the spurious concepts associated with each class, BEE provides actionable insights that can inform corrective strategies. For example, Section 4.5 shows that the spurious concepts revealed by BEE can be used as regularization signals, encouraging the model’s decision boundaries to stay equidistant from those biased directions, thereby improving worst-group accuracy across datasets. More broadly, BEE can complement ongoing mitigation approaches such as GroupDRO, counterfactual data generation, or fairness-aware fine-tuning, by supplying them with explicit and interpretable bias indicators derived from the model’s weight geometry.

---

### Official Review · Reviewer_5jdF · 2025-11-01

**Soundness:** 3
**Presentation:** 3
**Contribution:** 3
**Rating:** 6
**Confidence:** 3

**Summary:**

This manuscript introduces BEE (Bridging Explainability and Embeddings), a framework designed to diagnose spurious correlations learned during linear probing of frozen model. It uses linear probing to reveal spurious features that persist after full fine-tuning and transfer across diverse SOTA models. BEE is evaluated on vision (Waterbirds, CelebA, ImageNet-1k), language (CivilComments) and medical notes (MIMIC-CXR); and across embedding families (CLIP, CLIP-DataComp.XL, mGTE, BLIP2, SigLIP2). BEE surfaces concepts that can slash ImageNet accuracy by up to 95% and exposes clinical shortcuts in MIMIC-CXR dataset that cause false negatives.

**Strengths:**

1. Core idea is clear: both the linear head and concepts are in the same embedding space. Direct comparison of learned weights and concept embeddings constitutes a direct probe into the decision mechanisms of the classifier.
2. Efficient as it eliminates the need for expensive backbone retraining or the construction of elaborate counterexample data splits.
3. Empirical validation across multiple modalities / datasets and different foundational embedding families (e.g., CLIP and BLIP-2).
4. ImageNet-1k generative experiment is good evidence: prompting generator with BEE surfaced concepts leads to large drops in accuracy and even to class swaps.

**Weaknesses:**

1. Missing implementation details for reproducibility about the construction of the concept pool (prompts used with Llama-3.1-8B-Instruct, the exact filtering and de-duplication rules applied with WordNet and the final vocabulary size for each dataset).
2. BEE has a variable number of concept prompts per class. No details on whether the baselines like B2T / SpLiCE were given an equivalent prompt budget - difficult to ascertain if the observed gains are because of the higher quality of the BEE selected concepts or because of a larger query allowance.
3. Meaning of reversed diagonal reference, specification of the smoothing window and the exact roles & sensitivity of the r and p are not clearly defined. Sensitivity or ablation study is needed to confirm the stability of the concept selection process.
4. Missing empirical results / experiments scoring for negatively correlated concepts introduced in the paper.
5. Suggest adding multi-seed results with CIs for all tables.

**Questions:**

1. Could you please share epochs, batch sizes, learning rate, temperatures (for CLIP and non CLIP encoders), and hardware / compute used for each experiment?
2. Do you actually use negative SCs anywhere? If yes, where and how are they incorporated? If not, please explain why and provide an example showing whether negatives are useful.
3. Could you please provide the full procedure for building C_all?

---

> ### Author Response · Authors · 2025-11-20
>
> Dear Reviewer 5jdF
>
> We sincerely thank you for the thorough and constructive feedback. We greatly appreciate your recognition of BEE's strengths - the clarity of our core idea, the efficiency gains from eliminating the need for expensive training, and the breadth of our empirical validation across modalities and embedding families. We are equally grateful for your detailed identification of areas requiring improvement, such as the missing implementation details. We hope to address the raised questions and concerns through our responses below.
>
> ---
>
> > W1. Missing implementation details for reproducibility about the construction of the concept pool (prompts used with Llama-3.1-8B-Instruct, the exact filtering and de-duplication rules applied with WordNet and the final vocabulary size for each dataset).
>
> We have indeed omitted these details in the initial version of the paper. We listed below the required information and we will update the Appendix to include it.
>
> In the filtering stage we process a list of n-grams extracted from the set of image captions or from the set of samples in the case of CivilComments dataset, and output a list of strings that do not contain concepts related to the classes.
>
> We provide the prompt for the LLM in our response to W5 of Reviewer hqDR. We provide a set of examples to the LLM to improve its success rate, leveraging its in-context learning capabilities
>
> **Post-processing rules used in LLM-based filtering:**
> - We first locate the two apostrophes that are expected to be in the answer and keep only the part of the answer that lies between them, or a null string if two apostrophes are not present in the answer.
> - We ignore an n-gram if the answer of the LLM is a null string or if it contains words not in the initial sequence. This can signal a hallucination or failure to adhere to the required output format. We used the word tokenizer from nltk in this step.
> - For n-grams where the LLM provides a valid answer we trim the starting and trailing stopwords. We use the set of english stopwords in nltk, as well as the words “next” and “many”, which we found to be often produced by the captioning model.
> - If the answer of the LLM only contained stopwords we discard it. Otherwise it is added to the list of candidates.
>
> **Rules used in WordNet-based filtering:**
> - We first tokenize each n-gram into individual words with the word tokenizer in nltk.
> - We then lemmatize each word and mark it as class-related if its lemma appears in the class name (e.g., ”blonde” appears in “blonde hair” for CelebA) or if it is a hypernym or hyponym of the corresponding WordNet synsets for any class (e.g., “pelican” is a hyponym of “bird”). Note that the user could define one or more WordNet synsets for a class if it is a union of multiple synsets in WordNet, and a fitting hypernym does not exist. E.g., class 0 in a dataset could represent venomous animals, which includes certain snake species, as well as spiders, stingrays etc..
> - Each n-gram with no class-related words is added in the final list of candidates
> - For the other n-grams we remove all class-related words. If only stopwords remain, we discard them. Otherwise, the resulting string is added to the list of candidates.
>
> At the end of the filtering stage we deduplicate the obtained list of candidates. To be noted, we can have a word with both its singular and plural form in the list of candidates. The lemmatization in WordNet-based filtering is only used to determine if the word is related to the classes or not, but raw words are used in subsequent steps. This is an intended behavior because if only the plural form ever appears then we could gain additional information about the dataset - the entity always appears in a group.
>
> **Final number of candidates per dataset:**
> - Waterbirds: 294
> - CelebA: 280
> - CivilComments: 587
> - ImageNet: 368,170
>
> > W2. BEE has a variable number of concept prompts per class. No details on whether the baselines like B2T / SpLiCE were given an equivalent prompt budget - difficult to ascertain if the observed gains are because of the higher quality of the BEE selected concepts or because of a larger query allowance.
>
> We first argue that having more SC-candidates (and implicitly prompts) is not necessarily an advantage in the zero-shot classification experiment. Looking at the ablations in Table 8 from the Appendix, using a larger number of candidates (up to 30 per class) leads to worse performance. Simply using more prompts is thus not a shortcut for better results.
>
> We also want to clarify a detail of the experimental setup that was not explicitly mentioned in Section 4.1. Following B2T, each class has an equal prompt budget. For any class we build prompts using both its own SCs, as well as the SCs of the other class(es). We expand more on this topic in our response to **Q2&W4**.

---

> ### Author Response · Authors · 2025-11-20
>
> > W3. Meaning of reversed diagonal reference, specification of the smoothing window and the exact roles & sensitivity of the r and p are not clearly defined. Sensitivity or ablation study is needed to confirm the stability of the concept selection process.
>
> We agree that the “reversed diagonal” reference is not very clear. To simplify the description we will change the phrasing to a simpler and explicit form, referring to it as the “straight line connecting points (1, $s_{k,i}^+$) and (p, $s_{k,p}^+$)”.
>
> In our experiments we used a window of $r=5$ to smooth the scores of the candidates. The variable $p$ represents the number of points left after applying the smoothing with window $r$ over a sequence of $q$ score, only on the valid positions: $p=q-r+1$.
>
> To test the sensitivity of the concept selection w.r.t. the parameter $r$, we repeated the process for $r=3$ and $r=7$ on the Waterbirds dataset. In both cases the change of $r$ resulted in selecting 1, respectively 2 more keywords as SCs for one of the classes. Using the updated list of SCs in the SC-enhanced zero-shot prompting experiment resulted in minor differences (less than 0.1% change in average accuracy and 0.3% in worst group accuracy).
>
>
> > W5. Suggest adding multi-seed results with CIs for all tables.
>
> It is true that the only table featuring multi-seed results is Table 5. This is due to the fact that the mitigation procedures employed in Table 5 require training. In contrast, the zero-shot protocol employed in Tables 1 and 6 is deterministic. Similarly, the evaluations performed in Tables 3, 9, 10 and 11 are based on pretrained ImageNet models, and they are thus deterministic as well.
>
> > Q1. Could you please share epochs, batch sizes, learning rate, temperatures (for CLIP and non CLIP encoders), and hardware / compute used for each experiment?
>
> We list below all the hyper-parameters used in the training of the linear layers. We will also modify Section 3 to include any missing ones.
> - The number of epochs is not predefined. Training stops when the class-balanced validation accuracy has not improved for 5 epochs
> - AdamW optimizer
> - Batch size: 1024
> - Learning rate of 1e-4
> - Weight decay of 1e-5
> - Temperature: logits are multiplied by 100 for all encoders
> - GroupDRO $\eta$ : 1e-2
>
> In terms of hardware, we used an Nvidia RTX 4090 for captioning and image generation and an Nvidia RTX 2080 for linear probing.
>
> >**Q2&W4**    Usefulness of negative SCs
>
> We extend below on the implicit use of negative SCs in our experiment and we also did an additional experiment based on them.
> **Binary Classification Tasks (CelebA, Waterbirds, CivilComments).** In binary settings, a positive SC for one class implicitly serves as a negative SC for the other. A positive SC of class 0 favours the prediction of this class 0, implicitly hindering the prediction of class 1. Positive and negative SCs in a task are thus entangled - they share the same underlying pool of concepts.
> This is also reflected in the symmetry between the positive-SC score of class 0 and the negative-SC score for class 1, for the binary classification setting: $s_{0,i}^+ = s_{1,i}^-$. Proof:
>
> Let $ a_i = w_0^{\*\top} M(c_i) $ and $b_i = w_{1}^{*\top} M(c_i)$. We have that $s_{0,i}^+ = a_i - \text{min(}a_i, b_i\text{)}$ and $s_{1,i}^- = -b_i - \text{min(}-a_i, -b_i\text{)}$.
>
> Since $\text{min(}a_i, b_i\text{)} = a_i + b_i - \text{max(}a_i, b_i\text{)} = a_i + b_i + \text{min(}-a_i, -b_i\text{)}$, we have that $s_{0,i}^+ = a_i - \text{min(}a_i, b_i\text{)} = a_i - a_i - b_i - \text{min(}-a_i, -b_i\text{)} = -b_i - \text{min(}-a_i, -b_i\text{)} = s_{1,i}^-$.
>
> The SC-enhanced zero-shot classification application from B2T uses the positive SCs for both classes to create more expressive prompts. That is, for a class we use both its positive and negative SCs. We repeated the experiment, using only the positive or only the negative SCs identified by BEE on Waterbirds, leading to significant drops in performance:
>
>
> | Zero-Shot Prompting | Worst Acc % | Avg. Acc % |
> |---|--- | - |
> | with All SCs (in paper) | 50.3 | 86.3 |
> | only with Negative SCs | 34.7 | 60.9 |
> | only with Positive SCs | 16.7 | 67.9 |
>
>
> Intuitively, using only the positive SCs for a class amplifies the SCs (we are using prompts of waterbirds in water environments and landbirds in land environments), while using only the negative ones reverses the SCs.
>
>
> **Role of Negative SCs in generic settings.** Negative SCs of a class hint at the conditions in which said class is less likely to be correctly identified. As such, in a setting where generating and labeling new data points is feasible, the negative SCs indicate what concepts should be depicted near this class in order to mend the SCs learned from the original dataset.
>
>
> >**Q3**    Could you please provide the full procedure for building C_all?
>
> $C_{all}$ in Algorithm 1 is the output of the YAKE algorithm, a set of n-grams extracted from the captions of the dataset being studied.

---

### Official Review · Reviewer_hqDR · 2025-11-01

**Soundness:** 2
**Presentation:** 2
**Contribution:** 2
**Rating:** 4
**Confidence:** 4

**Summary:**

The paper introduces BEE (Bridging Explainability and Embeddings), a framework for identifying spurious correlations by analyzing model weight space and embedding geometry rather than relying solely on data splits or prediction errors. BEE examines how fine-tuning perturbs pretrained representations and uses linear probing as a transparent diagnostic tool to reveal hidden shortcuts. It demonstrates that many spurious features persist even after fine-tuning and can transfer across diverse state-of-the-art models and domains, including vision and language tasks. Experiments show that BEE effectively uncovers severe and transferable spurious correlations, establishing it as a principled framework for model auditing and trustworthiness.

**Strengths:**

1. The authors showcase a wide range of use cases, including zero-shot classification, discovering spurious correlations within datasets, and applications to both text-based and image-based datasets.

**Weaknesses:**

1. The proposed method closely resembles the concepts of Label-free CBM [1] and Post-hoc CBM [2], both of which also utilize CLIP-based image /text encoders. In particular, Post-hoc CBM (see Table 10 in its Appendix) demonstrates a similar approach to identify biased concepts residing in the dataset. While the authors could emphasize the spurious concept discovery component as their main contribution, the process of using LLMs and captioning models to enumerate and filter potential concepts appears similar to LLaVA-CBM [3]. The paper should more clearly articulate its unique contribution and novelty relative to these existing works.

2. What would happen if the encoder component were fine-tuned rather than kept frozen?

3. The Method section requires more detail. Some crucial explanations are deferred to Section 3, making it difficult to follow the approach before reaching the experiments. Including a concrete example using a well-known dataset could substantially improve clarity.

4. Section 4.5 (Experiment Setup) is difficult to understand. The authors should provide a clearer description, particularly regarding the setup and implementation details of GroupDRO.

5. Since the performance of the image captioning model appears to play a crucial role, it would be helpful to show example outputs and explain how LLMs are used (e.g., prompts, output format, etc.). Have the authors experimented with different captioning or language models, and if so, how do the results vary?

6. **Writing:** Some sentences are overly long and contain excessive commas, which makes them difficult to read.

[1] Label-Free Concept Bottleneck Models, ICLR 2023

[2] Post-hoc Concept Bottleneck Models, ICLR 2023

[3] Constructing Concept-based Models to Mitigate Spurious Correlations with Minimal Human Effort, ECCV 2024.

**Questions:**

1. Section 4.5 (Experiment Setup) is unclear and requires further clarification. What is the exact setup for GroupDRO? Did you use the same CLIP-based encoder and train only the classifier? Were all worst-group samples removed from the original training set. And if so, what is the rationale behind this choice? In that case, labeling the baseline as “GroupDRO” may not be appropriate, as the setup differs from the original formulation. The authors should provide a detailed justification for this experimental design.

2. In Table 5, the improvement on the CelebA dataset appears noticeably smaller compared to other datasets. Could the authors explain the reason behind this limited gain?

---

> ### Author Response · Authors · 2025-11-20
>
> Dear Reviewer hqDR, we thank you for your thoughtful feedback and for recognizing BEE's broad applicability across vision and language tasks. We appreciate your questions regarding our positioning relative to Label-free CBM, Post-hoc CBM, and LLaVA-CBM. We are grateful for your suggestions to improve clarity, particularly regarding the Method section’s structure and the GroupDRO experimental setup in Section 4.5. We believe the presentation from our manuscript will be greatly improved by incorporating your feedback.

---

> ### Author Response · Authors · 2025-11-20
>
> > **W1.** Comparison with the CBM literature.
>
> We currently discuss our relationship to CBMs (including references to Oikarinen et al. \[1\]) in Appendix E. Given the additional page, we will move the discussion from Appendix E into the main paper and further enrich it.
>
> **General similarities and differences:** In terms of similarities to the overall CBM literature, as you have evidentiated, spurious correlations can be revealed using both our method and CBMs. However, our proposed method and line of work differs from CBMs in significant ways, both in terms of goals and in terms of the effects upon the models’ performance. The CBM literature in general aims at proposing and training explainable-by-design layers. That is, given a set of $M$ concepts, a bottleneck layer is trained to create a new representation $c \\in \\mathcal{R}^M$ where each dimension measures the presence or absence of a concept from the predefined list. This representation is used in downstream tasks and offers a high degree of interpretability, but may potentially be less expressive and less performant than the original representation. Given that the features of this bottleneck layer are interpretable, spurious correlations can be found by means of analyzing how these features are used for the downstream task. In contrast to this approach, our aim is to investigate spurious correlations learned by current state-of-the-art models during their unaltered training procedures. That is, we aim to uncover SCs learned by general state-of-the-art models used in the community, which are not explainable by design. Overall, our approach and CBMs offer a different tradeoff between explainability and expressivity.
>
> **Oikarinen et al. \[1\]** Similar to our approach, the method proposed by Oikarinen et al. \[1\] can be applied in general setups, but with some caveats. Oikarinen et al. \[1\], train a concept layer, within which each neuron signals the presence or absence of a predetermined concept proposed by GPT-3. This intervention constrains the reasoning space of the model down to the set of predetermined concepts, yielding, compared to unaltered models, drops in accuracy of up to 4.97%, as reported in Table 2 from Oikarinen et al. \[1\]. Different from Oikarinen et al. \[1\], we never constrain the model in any way, shape or form. Our analysis is completely post-hoc, allowing the model to reason in its natural embedding space while achieving the best performance it can on the given dataset.
>
> **Yuksekgonul et al. \[2\]** Similar to our method, the approach proposed by Yuksekgonul et al. \[2\] is post-hoc. That is, the authors train a CBM layer on top of a pre-trained non-CBM model. Furthermore, their approach features a mechanism designed to reduce the performance loss usually seen in CBM models. In this respect their work differs significantly from prior CBM literature. Our work differs from this approach in two main respects pointed out directly by Yuksekgonul et al. \[2\] in their “Limitations and Conclusion” section of their paper: “Users should be careful about the concept dataset used to learn concepts, which can reflect various biases. While there are several such real-world tasks, it is an open question if human-constructed concept bottlenecks can solve larger-scale tasks(e.g. ImageNet level).” First, in this direct quote, the authors point out that their method cannot find out the spurious correlations learned by their bottleneck layer during training. In contrast, our method is specifically designed to point out the spurious correlations learned by our classifier during training. Second, while in the work of Yuksekgonul et al. \[2\] the possibility of applying their method to large-scale datasets such as ImageNet remained an open question, we have successfully applied our method on the ImageNet dataset.
>
> **Kim et al. \[3\]** Similar to our method, the approach proposed by Kim at al. \[3\] does employ LLMs to propose and filter concepts. Their work further improves upon methods such as the one proposed by Yuksekgonul et al. \[2\] by means of proposing a mechanism meant to automatically obtain unbiased MLLM attribute annotations in order to train the bottleneck layer without human supervision. From this standpoint, out of the works discussed here in particular, this approach stands out as the most refined and most similar to ours. In terms of differences, even if the MLLM annotations were to match the human level, the limitations pointed out by Yuksekgonul et al. \[2\] still remain. That is, the spurious correlations learned by the bottleneck layer itself during training remain hidden. The annotation procedure ensures that the labels that are used are as accurate as possible, but it cannot determine what spurious correlations are present in the training dataset used for the bottleneck layer. In contrast, our method investigates the spurious correlations learned by our classifier during training.

---

> ### Author Response · Authors · 2025-11-20
>
> > **W2.** What would happen if the encoder component were fine-tuned rather than kept frozen?
>
> The classification datasets employed in this area of the literature, including ImageNet, are 400 (ImageNet) to 400,000 (Waterbirds) times smaller than the dataset used to train CLIP-ViT-B-16. Fine-tuning would thus cripple the model and would result in losing a lot of the pretrained concepts. However, unfreezing the encoder is not necessary. We point out that our linear probe is a diagnosis tool meant to highlight the spurious correlations that are learnable from a given dataset. Since these spurious correlations plague the dataset, any model trained on the investigated dataset will be affected by them to one degree or another. This can be seen in our results from the ImageNet experiments in Tables 3, 9, 10 and 11, where we have tested state-of-the-art models fully trained on ImageNet showing the degree to which they are affected. Thus, our results already generalize to an extensive set of fully trained models. Furthermore, keeping the encoder frozen enables us to perform the weight space analysis.
>
> During training, we deliberately keep the CLIP image and text encoders frozen to ensure that all image and all text encodings lie at all times in the same shared latent space of the pretrained CLIP. Furthermore, towards this end, we initialize the weights of our linear probe with the embedding of the class name for each class and we do not use a bias. That is, initially, the weight used for the Fire Truck class corresponds exactly to and lives in the same space as the embedding yielded by the text encoder for the concept ‘Fire Truck’. During training, we only optimize this weight while ensuring that it remains within the same shared latent space. To enforce this, we follow the exact training procedure proposed by OpenAI, providing the training code as an anonymous repository in Appendix C. We keep our class embeddings/weights normalized at all times, we compute the cosine similarity between them and the input images and we compute the final loss as a temperature-scaled softmax on top of these similarities. Thus, our weight begins as the embedding of a concept (the textual description of the class) and it is trained to exhibit a high cosine similarity with the image embeddings (which live in the same shared latent space) from the class (just as in the original CLIP training procedure). In the end, given the training procedure and the fact that we keep the text and image encoders frozen, the weight lives in the same exact latent space in which it started and it stands as an optimized text embedding. This is the procedure that enabled us to perform the weight-space spurious correlation analysis.
>
> To summarize, the spurious correlations detected by our approach already generalize to fully finetuned models and keeping the encoder frozen was a deliberate choice in our diagnosis protocol.
>
> > **W3.** The Method section requires more detail. Some crucial explanations are deferred to Section 3, making it difficult to follow the approach before reaching the experiments. Including a concrete example using a well-known dataset could substantially improve clarity.
>
> Thank you for providing us with this feedback. We will enhance the presentation from the manuscript with an example in order to improve clarity.

---

> ### Author Response · Authors · 2025-11-20
>
> >  **W4.** Section 4.5 (Experiment Setup) is difficult to understand. The authors should provide a clearer description, particularly regarding the setup and implementation details of GroupDRO.
> > **Q1.** Section 4.5 (Experiment Setup) is unclear and requires further clarification. What is the exact setup for GroupDRO? Did you use the same CLIP-based encoder and train only the classifier? Were all worst-group samples removed from the original training set. And if so, what is the rationale behind this choice? In that case, labeling the baseline as “GroupDRO” may not be appropriate, as the setup differs from the original formulation. The authors should provide a detailed justification for this experimental design.
>
> The experiment described in Section 4.5 was specifically designed to distinguish our method from the subpopulation shift literature and it indeed features an altered dataset. In the subpopulation shift literature \[Change is Hard\], even though classes are spuriously correlated with specific concepts, all classes feature bias-conflicting samples. The approaches within this domain thus generally rely on upsampling or upweighting these bias-conflicting samples. GroupDRO, the most popular approach, dynamically weights groups of samples during training based on the magnitude of the loss, emphasizing harder samples. Methods designed to find spurious correlations within subpopulation shift setups \[xrm, feed\] crucially rely upon two assumptions: (i) that the training dataset already features counterexamples able to point out spurious correlations, and (ii) that the dataset can be conveniently partitioned into (class, concept) groups. As such, they are not applicable to generic datasets such as ImageNet, which cannot conveniently be grouped into (class, concept) partitions. Furthermore, ImageNet also lacks counterexamples for some spurious correlations. For instance, crayons appear in pictures with erasers, and they become spuriously correlated, but there are no pictures in ImageNet with crayons near animals that can be misclassified by the model in order to reveal and help correct the spurious correlation. As such, there are no samples that can be upweighted by GroupDRO. By performing experiments on ImageNet we show that our method transcends the current approaches from the subpopulation shift literature, by means of being applicable in generic setups lacking counterexamples.
>
> However, the fact that ImageNet does not have attribute and spurious correlation annotations prevented us from performing a quantitative evaluation. As such, we have tried to replicate the fully spurious setup, lacking counterexamples, using the annotated datasets that are popular in the subpopulation shift literature. Thus, we have removed the counterexamples from the datasets mentioned in Table 5, as you have noted. For each class, we have only kept the samples that contained the feature which was spuriously correlated with the class. For instance, in the Waterbirds setups, we have only kept images featuring waterbirds with water backgrounds and landbirds with land backgrounds. In the case of CelebA, we have only kept images of blonde women and dark haired men. These changes affect only the training and validation datasets, the test dataset remained untouched in order to validate how much the models are affected by the spurious correlations. Using this setup we were able to point out that, different from existing literature, our method is able to find spurious correlations even when there are no counterexamples in the training dataset. GroupDRO is applicable in this setup, but since there is only one group per class, the groups are equivalent to the classes and thus GroupDRO serves only to balance classes. Different from GroupDRO, our proposed regularization together with the spurious correlations detected by BEE are able to improve the performance in this setup.

---

> ### Author Response · Authors · 2025-11-20
>
> > **W5.** Since the performance of the image captioning model appears to play a crucial role, it would be helpful to show example outputs and explain how LLMs are used (e.g., prompts, output format, etc.). Have the authors experimented with different captioning or language models, and if so, how do the results vary?
>
> To answer this question we have employed a different LLM, namely GPT-3.5-turbo for the Waterbirds zero-shot experiment presented in Table 1\. We further provide: (i) the prompt used for both the original LLM (Llama 3 8B) and the new one (GPT-3.5-turbo), (ii) a qualitative comparison between the outputs of the two models, and (iii) a quantitative comparison in terms of the zero-shot performance improvement brought by employing the SCs found.
>
> **Prompt used for both Llama and GPT:**
> ```I will provide a list of concepts and sequence of words. Your task is to remove any instance of the concepts from the given sequence.
> If no instance of any concept is present then you must return the sequence as is.
> Here are a few examples:
> Example 1:
> Concepts: [dogs and any specific species of dogs]
> Sequence: 'a golden retriever with a bone'
> Answer: 'bone'
>
> Example 2:
> Concepts: [clothing and anything related to their color]
> Sequence: 'a shiny black and white dress'
> Answer: 'shiny'
>
> Example 3:
> Concepts: [mentions of people's names]
> Sequence: 'John is an assistant'
> Answer: 'assistant'
>
> Example 4:
> Concepts: [cats, horses, dolls, the sun and any specific species or types of these concepts]
> Sequence: 'A picture of the rising sun'
> Answer: 'picture'
>
> Now complete the following case, without thinking step by step or asking for anything else.
> Concepts: [{}]
> Sequence: '{}'
> Answer:
> ```
>
> **Qualitative comparison of filtering done by Llama and GPT**
>
> | Original n-gram | Llama-3.1 output | GPT-3.5-Turbo output |
> |-----|------|-----|
> | bird flying over a body | flying over a body | flying over a body  |
> | black bird sitting on top | sitting on top | sitting on top  |
> | yellow and black bird standing | standing | yellow and black standing  |
> |bird in a bamboo forest | bamboo forest | in a bamboo forest  |
> | bird sitting on a boat | boat |  sitting on a boat  |
> | flying over the ocean | flying over the ocean | ocean  |
> | black bird sitting | sitting | sitting |
>
> **Performance comparison (Llama vs GPT):**
> We have reproduced the zero-shot experiments presented in Table 1\. We have noted a 0.8\% decrease in terms of Worst Group Accuracy (WGA), from 50.3\% (Llama) $\\rightarrow$ 49.5\% (GPT). As such, the results do indeed differ based on LLM’s influence, but not in a make or break fashion.
>
> > **W6.** Writing: Some sentences are overly long and contain excessive commas, which makes them difficult to read.
>
> Thank you for providing us with this feedback, we will restructure long phrases in order to make the text more fluid.
>
> > **Q2.** In Table 5, the improvement on the CelebA dataset appears noticeably smaller compared to other datasets. Could the authors explain the reason behind this limited gain?
>
> For this experiment the SCs found using our method match the known profile of the dataset. The fact that these SCs were found but not mitigated implies that the regularization employed is not strong enough to aid sufficiently in this context.
>
> ---
>
> We thank you for the suggestions and we believe that incorporating these changes into the manuscript will greatly improve the presentation of our work.

---

### Author Response · Authors · 2025-12-03
**Summary for the Area Chair**

We thank the reviewers for their time and effort in evaluating our submission, even though they cannot provide further comments given the circumstances. To assist the AC, we provide below a consolidated summary of our responses to the main points raised.




## **Relation to Concept Bottleneck Models (CBMs)**
First, we note that the connection to CBMs, including several references mentioned by reviewer hqDR, was already discussed in Appendix D of the original manuscript (we have now expanded the discussions into the main text). Our paper benefits from this connection to the extensive CBM literature, as we share similar goals of achieving interpretability and robustness through concepts. However, as we have explained, our approach operates differently from CBMs and offers advantages not present in prior work.
We study the spurious correlations learned by state-of-the-art models without introducing a bottleneck representation; the model’s dense features remain unchanged. This allows us to gain conceptual insights and a degree of interpretability while preserving the full representational power of the original features.
Overall, our method and CBMs reflect different trade-offs between interpretability and representational capacity, and each can be used in complementary ways to gain insights and improve model robustness.


## **Are Linear Probes Sufficient for Analyzing Spurious Correlations?**
We purposefully chose linear probes to extract the concepts since they offer us interpretability and the choice is strongly supported by prior work: Kirichenko et al. (ICLR 2023) show that core and spurious features are typically linearly separable in final-layer representations. Recent work using Sparse Autoencoders (Huben et al., ICLR 2024) demonstrates that many meaningful, disentangled features can be isolated through a single linear transformation. Given this evidence, linear probing offers an effective balance between interpretability and representation power for identifying SCs in final-layer embeddings.


## **Computational Overhead**
Preprocessing is fully automated and critical for SC quality. For ImageNet-1k, the most expensive dataset to preprocess, it takes ~12 hours on an RTX 4090, incurred only once per dataset. Afterward, models can be evaluated efficiently, with linear probing taking ~2 minutes per model.


**Manuscript revision**: We have integrated the feedback from the reviewers into the manuscript to clarify the topics discussed during the rebuttal phase and to improve readability, comparability, and reproducibility. Code has been available since the initial submission, and updates are marked in red in the pdf.

---

### Meta-Review · Area_Chair_DKJa · 2026-01-10

**Summary:**

This paper received all slightly positive ratings except one (rev. hqDR), overall the scores are 4, 6, 6, 6.

Main criticisms regard the weak novelty (especially wrt to a couple of works), and especially the lack of method's details and requests of clarifications of several parts (e.g., number of concepts per class, meaning of reversed diagonal reference, etc.), but also some issues at a more general level are raised (e.g., effect of the foundation model vs. task/dataset used, possible bias).

Also the experimental part is criticised, namely, experimental design difficult to understand and lack of implementation details here, insufficient ablations, missing explainable examples, results not well commented. It's also unclear how LLM are used in image captioning task, and paper presentation is considered a problem.

**Reviewer Concerns:**

The feedback provided to each review by the authors considers all raised aspects. All issues are discussed but it is unclear if well addressed since the complete lack of discussion does not help such verification.

To my opinion, novelty issue is partially fixed since doubts remain especially wrt [3]: The problems are about the a-priori definition of the the concepts, on the use of datasets to learn the concepts which can have bias, on the training of the concepts in general, and how to mitigate SCs during the training.

[3] Constructing Concept-based Models to Mitigate Spurious Correlations with Minimal Human Effort, ECCV 2024.

**Reviewer Scores:**

Overall, the majority of the scores are positive, but the number and severity of the comments deserve an accurate revision of the paper.

The paper results revised especially in the supplementary material. The revision on the main paper is present but does not change much the overall presentation as it would be required. Only Sect. 5 is added ex-novo providing a useful discussion of the results.

It is recommended that in the CR ready version, the paper can include the information requested by the reviewers regarding the methodology, the experimental protocol, as well as the novelty of the approach.

---

### Decision · Program_Chairs · 2026-01-26

Accept (Poster)